# ToolRL: Reward is All Tool Learning Needs

**Cheng Qian, Emre Can Acikgoz, Qi He, Hongru Wang, Xiusi Chen,**
**Dilek Hakkani-Tür, Gokhan Tur, Heng Ji**

University of Illinois Urbana-Champaign
{chengq9,acikgoz2,hengji}@illinois.edu

## Abstract

Current Large Language Models (LLMs) often undergo supervised fine-tuning (SFT) to acquire tool use capabilities. However, SFT struggles to generalize to unfamiliar or complex tool use scenarios. Recent advancements in reinforcement learning (RL), particularly with R1-like models, have demonstrated promising reasoning and generalization abilities. Yet, reward design for tool use presents unique challenges: multiple tools may be invoked with diverse parameters, and coarse-grained reward signals, such as answer matching, fail to offer the finegrained feedback required for effective learning. In this work, we present the first comprehensive study on reward design for tool selection and application tasks within the RL paradigm. We systematically explore a wide range of reward strategies, analyzing their types, scales, granularity, and temporal dynamics. Building on these insights, we propose a principled reward design tailored for tool use tasks and apply it to train LLMs using RL methods. Empirical evaluations across diverse benchmarks demonstrate that our approach yields robust, scalable, and stable training, achieving a 17% improvement over base models and a 15% gain over SFT models. These results highlight the critical role of thoughtful reward design in enhancing the tool use capabilities and generalization performance of LLMs. All the codes are released to facilitate future research.[1]

## 1 Introduction

Recent advances in Large Language Models (LLMs) have showcased remarkable capabilities in complex reasoning tasks [20]. Among the techniques that have significantly contributed to this progress, Reinforcement Learning (RL) has emerged as a powerful paradigm, enabling LLMs to develop emergent capabilities such as self-reflection, self-correction, and long-horizon planning [13, 48]. These capabilities have been instrumental in the success of models like o1 and R1, particularly in mathematical and logical reasoning domains [36, 15, 23, 18].

Beyond traditional reasoning tasks, an increasingly important area is **Tool-Integrated Reasoning** (TIR). TIR involves LLMs interacting with external tools, such as search engines [17, 62], calculators [4, 37], or code interpreters [12, 24], in a multi-step, feedback-driven loop to arrive at solutions. TIR is particularly important as it addresses core limitations of LLMs, such as outdated knowledge and calculation inaccuracy. By integrating external tools that offer real-time access and specialized capabilities, TIR enables models to tackle complex tasks in a more grounded and goal-directed way.

Unlike textual reasoning, which primarily involves deduction and inference from static text, TIR additionally demands the model's ability to select appropriate tools, interpret intermediate outputs, and adaptively refine its trajectory on the fly. These dynamic and interactive reasoning skills position TIR at the core of the emerging paradigm of LLMs-as-agents. As such, TIR enables a wide range

---

[1] Data and codes released at https://github.com/qiancheng0/ToolRL

of applications, including scientific discovery [41, 16], research automation [2, 51], embodied task completion [60, 14], and everyday decision-making [54, 59].

Training LLMs for TIR tasks has predominantly relied on Supervised Fine-Tuning (SFT), wherein existing approaches typically generate these integrated reasoning steps offline, followed by subsequent SFT on these trajectories [3, 58, 8, 1]. While SFT is effective to some extent, it struggles with generalization, exploration, and adaptability [9, 13], failing to capture the strategic flexibility needed for optimal tool use. This motivates a fundamental research question: *Can RL better equip LLMs with agentic tool-using capabilities, and if so, what is the optimal RL design for TIR?*

Recent efforts such as Search-R1 [17] and TORL [23] have begun to explore this direction. However, their focus is narrow: either constrained to search tools in question answering settings or code tools in math problem-solving. In contrast, our work aims to study RL-based training for *general-purpose* tool selection and application, across diverse and complex tool sets with different task types.

For an RL algorithm to be effective, a well-designed **reward** is essential. Unlike math tasks with a single correct answer, Tool-Integrated Reasoning (TIR) tasks introduce multiple layers of complexity: they often involve multi-step interactions where each turn may require invoking multiple tools, each with carefully specified parameters. Designing effective reward signals to guide learning through this complexity remains an open and underexplored challenge. In this paper, we address reward design for TIR by proposing a framework applicable across RL algorithms. We validate it on Group Relative Policy Optimization (GRPO) [44] and Proximal Policy Optimization (PPO) [43], highlighting its effectiveness in improving tool use under GRPO.

We begin by formalizing the TIR task, and outlining general principles for effective reward design. Building on this foundation, we show how RL algorithm can be leveraged to train LLMs for robust and context-aware tool selection and application. Empirical results demonstrate that our approach outperforms base models by 17% and SFT models by 15% across multiple tool use and QA benchmarks. Moreover, the trained model exhibits strong generalization to unseen scenarios and task objectives, along with emergent behaviors such as proactiveness and metacognitive reasoning.

To identify optimal reward strategies, we systematically explore a broad spectrum of reward configurations across four key dimensions: (1) *reward type* (what aspect to reward), (2) *reward scale* (how much to reward), (3) *reward granularity* (how detailed the reward signal is), and (4) *reward dynamics* (how rewards evolve over time). Through extensive experiments, we identify reward designs that best align with agentic tool use and uncover insights into what makes a reward "useful" for tool invoking LLMs. We summarize the core insights we derive as follows:

- Longer reasoning trace is not inherently better and length rewards can degrade performance.
- Dynamic reward scale helps models transition smoothly from simple to complex behaviors.
- Finegrained reward decomposition leads to more stable and effective learning.

We summarize the overall contributions of our paper as follows:

- We present the first systematic study on RL-based training for general-purpose tool selection and application in LLMs.
- We propose a principled reward design framework for TIR and validate its effectiveness across RL algorithms, with particular strength demonstrated on GRPO.
- We conduct extensive experiments analyzing the effects of various reward strategies and distill actionable insights for future research on LLM-agent training.

## 2 Method

SFT often suffers from overfitting to certain patterns and constrains the model's ability to learn optimal strategies for tool use. To address this, we introduce a RL approach for enhancing TIR in LLMs. In this section, we begin by defining the TIR task (Section 2.1), followed by our customized rollout strategy (Section 2.2) and reward design (Section 2.3). These components are then integrated into the RL framework [44] to guide model training on general TIR tasks (Section 2.4).

### 2.1 Task Definition

*Tool-Integrated Reasoning* is the process of incorporating external tools into the reasoning trajectory of an LLM to solve a user task. A typical TIR trajectory involves multiple tool invocations over several

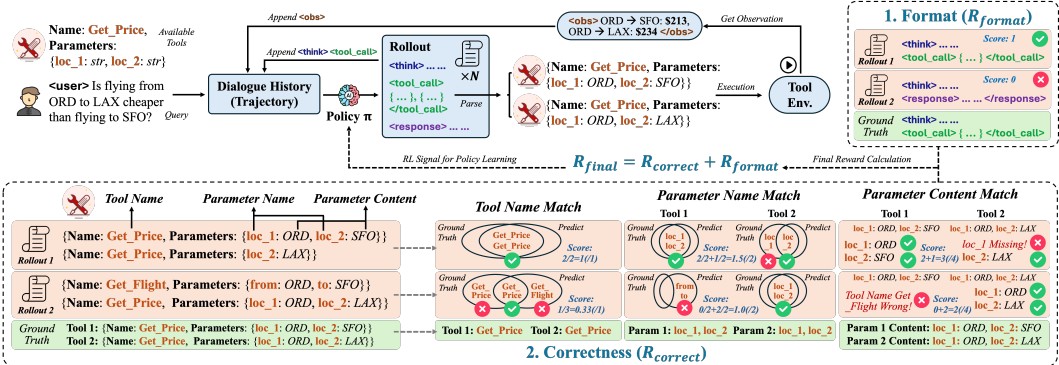

Figure 1: Illustration of TIR rollout and calculation of format and correctness reward.

reasoning steps, with the final outcome determined by the cumulative success of these intermediate decisions. Formally, given a tool set $\mathcal{T} = \{t_1, t_2, \ldots, t_n\}$ containing $n$ available tools, and a user query $Q$, the reasoning trajectory up to step $k$ is denoted as:

$$s_k = (r_1, \mathcal{T}_1, o_1), (r_2, \mathcal{T}_2, o_2), \ldots, (r_k, \mathcal{T}_k, o_k),$$

where $r_i$ denotes the model's natural language reasoning at step $i$, $\mathcal{T}_i \subseteq \mathcal{T}$ denotes the set of tool calls invoked at step $i$, and $o_i$ denotes the observation received after executing tools in $\mathcal{T}_i$, possibly including both environment and user feedback.

At each step $k + 1$, the model must generate the next reasoning step $r_{k+1}$, select a set of tools $\mathcal{T}_{k+1} \subseteq \mathcal{T}$, and formulate a grounded tool call (i.e., a parameterized invocation of each tool) to make progress toward solving $Q$. The model's policy is defined as $\pi : s_k \to (r_{k+1}, \mathcal{T}_{k+1})$, where the model's objective at each step is to select a tool set $\mathcal{T}_{k+1}$ that maximizes the reward:

$$\mathcal{T}_{k+1}^* = \arg \max_{\mathcal{T}_{k+1} \subseteq \mathcal{T}} R(s_k, \mathcal{T}_{k+1}, o_{k+1}),$$

where $R(\cdot)$ represents the reward function that evaluates progress made by invoking the tools in $\mathcal{T}_{k+1}$. While the immediate reward at each step is maximized, the model's policy is implicitly optimized to maximize the cumulative reward over the entire trajectory, formulated as:

$$\max_\pi \ \mathbb{E}_\pi \left[ \sum_{k=1}^{K} R(s_k, \mathcal{T}_{k+1}, o_{k+1}) \right],$$

This formulation is valid because our training data includes ground truth tool calls at each step, allowing step-wise reward signals to guide multi-step success. Unlike QA tasks that focus solely on the final answer, tool selection and application tasks provide dense intermediate feedback. Moreover, we later demonstrate that our method enables the model to generalize to settings where tool calls are free-form and only the final outcome matters. Therefore, our task setting encourages the model to optimize tool use at each step while aligning with the overall task goal.

## 2.2 TIR Rollout

To enable the model to autonomously generate reasoning traces and tool calls, we instruct the LLM to use special tokens <think>, <tool_call>, and <response> to indicate their thoughts, tool calls and responses in output. The full prompt content is shown in Figure 8.

As illustrated in Figure 1, when the model output includes <tool_call>, we automatically parse the tool calls into individual invocations using the model-predicted parameters. The outputs from executions are then inserted into the <obs> field and appended to the dialogue history, serving as the model's interaction trajectory. Similarly, if the output contains <response>, the corresponding response is parsed and appended to the dialogue history. Please refer to Figure 8 and Figure 9 in the Appendix for instruction details.

It is important to note that <tool_call> and <response> are not mutually exclusive; they may co-occur within a single output. The user's initial query $Q$ is placed in the *Initial User Input* placeholder, and any subsequent user inputs are also appended to the dialogue history when present.

## 2.3 Reward Design

Rule-based reward mechanisms have demonstrated strong empirical performance and are commonly employed. In our training, we similarly adopt a reward formulation that combines structural and correctness-based components [17, 23, 52]. Specifically, the format reward assesses whether the model output adheres to the expected structure including thoughts, tool calls, and responses, while the correctness reward evaluates the accuracy of tool invocations. Formally, the overall reward $R_{\text{final}}(\cdot)$ is decomposed into two components: $R_{\text{format}} + R_{\text{correct}}$, each described in detail below:

**Format Reward.** The format reward $\mathcal{R}_{\text{format}}$ checks whether the model output contains all required special tokens in the correct order as specified by the ground truth:

$$\mathcal{R}_{\text{format}} = 1 \;\; \textit{if} \;\; \text{all required fields appear and are in the correct order} \;\; \textit{else} \;\; 0$$

**Correctness Reward.** The correctness reward $\mathcal{R}_{\text{correct}}$ evaluates predicted tool calls $P = \{P_1, ..., P_m\}$ against ground-truth calls $G = \{G_1, ..., G_n\}$. It includes three components:

- *Tool Name Matching:*

$$r_{\text{name}} = \frac{|N_G \cap N_P|}{|N_G \cup N_P|} \in [0, 1]$$

  where $N_G$ and $N_P$ are sets of tool names extracted from the ground-truth and predicted tool calls.
- *Parameter Name Matching:*

$$r_{\text{param}} = \sum_{G_j \in G} \frac{|\text{keys}(P_G) \cap \text{keys}(P_P)|}{|\text{keys}(P_G) \cup \text{keys}(P_P)|} \in [0, |G|]$$

  where $\text{keys}(P_G)$ and $\text{keys}(P_P)$ are parameter names of ground-truth and predicted tool calls.
- *Parameter Content Matching:*

$$r_{\text{value}} = \sum_{G_j \in G} \sum_{k \in \text{keys}(G_j)} \mathbb{1}[P_G[k] = P_P[k]] \in [0, \sum_{G_j \in G} |\text{keys}(G_j)|]$$

  where $P_G[k]$ and $P_P[k]$ represent parameter contents of ground-truth and predicted tool calls.

The total match score for each match is derived as $r_{\text{match}} = r_{\text{name}} + r_{\text{param}} + r_{\text{value}} \in [0, S_{\text{max}}]$, where $S_{\text{max}} = 1 + |G| + \sum_{G_j \in G} |\text{keys}(G_j)|$ denotes the maximum possible score. The total score is computed by finding the optimal matching between $P$ and $G$ to maximize the total match score:

$$\mathcal{R}_{\text{correct}} = 2R_{\text{max}} \cdot \frac{r_{\text{match}}}{S_{\text{max}}} - R_{\text{max}} \in [-R_{\text{max}}, R_{\text{max}}]$$

where $R_{\text{max}}$ denotes the maximum possible score after normalization, for which we empirically set to 3 in all experiments. More analysis and ablations of reward scale is presented in Section 4

The final reward value $\mathcal{R}_{\text{final}}$ is finally derived as the sum of $\mathcal{R}_{\text{format}}$ and $\mathcal{R}_{\text{correct}}$. Unlike prior works that often rely on binary or overly simplified reward signals, our design captures the nuanced structure of tool calls by evaluating multiple interdependent components including tool names, parameter schemas, and parameter values. This finegrained formulation better reflects the complexity of real-world tool use, where correctness cannot be reduced to a single binary criterion. We further validate the impact of this design through comprehensive analysis in Section 4.

Overall, our reward design ensures a balanced and interpretable evaluation signal by explicitly separating structural compliance from semantic correctness. By aligning rewards with both format adherence and finegrained tool call accuracy, the model is guided to produce outputs that are not only syntactically valid but also semantically faithful, which is crucial for final task success.

## 2.4 RL Training

To tune the model with structured rewards, we apply both PPO and GRPO algorithms and employ the latter one as our main experiment setting. The details about GRPO algorithm tailored with TIR tasks is presented in Appendix C.

Unlike the original GRPO formulations, we omit the KL penalty term against a reference model. This design choice encourages the model to more freely adapt its behavior to our custom response format and structured reward signals. In practice, we observe that this leads to faster convergence and comparable performance, while also simplifying the training pipeline.

## 3 Experiments

### 3.1 Training Dataset

To support robust tool learning, we construct a mixed dataset spanning diverse tool use scenarios:

- **ToolACE** [27]: A general tool use dataset where the model learns when to invoke tools versus respond directly, improving decision-making in multi-step interactions.
- **Hammer (Masked)** [25]: A subset of Hammer with randomized tool and parameter names, forcing the model to rely on descriptions rather than memorized labels, thus enhancing generalization and reducing overfitting to certain tools.
- **xLAM** [61]: A compositional dataset requiring one or multiple tool calls per turn, encouraging the model to reason about tool dependencies and plan diverse tool calling action actively.

Empirically, we sample 2K examples from ToolACE and 1K each from Hammer and xLAM, creating a balanced dataset spanning diverse levels of complexity and tool use. Multi-step trajectories are decomposed into single-step instances, with prior dialogue history injected into the user prompt to preserve context. This encourages strategic exploration and teaches the model to apply tools appropriately within each step.

### 3.2 Experiment Settings

**Training.** We conduct all RL experiments using the veRL framework [46]. For each training step, we sample a batch of 512, and generate 4 responses per query, training for 15 epochs in total (see Appendix E for full configuration). To encourage policy exploration, we remove KL regularization and apply temperature 1.0. We initialize our models with the Qwen-2.5-Instruct [49] and Llama-3.2-Instruct [11] series, which are further tuned under our customized reward design.

**Evaluation.** We evaluate our approach on the **Berkeley Function Call Leaderboard** (BFCL) [30], a comprehensive benchmark that spans a diverse set of challenges, including single-step reasoning, multi-step tool use, real-time execution, irrelevant tool rejection, simultaneous multi-tool selection, and multi-tool application[2]. In addition, we present results on **API-Bank** [21], a three-level evaluation framework comprising 73 diverse and complex API tools. It assesses an LLM's ability to select and apply tools through natural multi-turn dialogues, across three levels of difficulty. We also evaluate on a representative QA benchmark **Bamboogle** [31], which comprises a variety of question-answering tasks where performance is measured based on the final answer accuracy rather than the correctness of tool use. These broad coverage makes our evaluation setting effective for evaluating real-world LLM tool use proficiency. All results are reported in terms of accuracy.

**Baselines.** We set GRPO cold start as our main setting and compare it against several baselines: (1) **Raw Instruct Model**: the original model without any additional fine-tuning or RL. (2) **SFT on RL Data**: the instruct model fine-tuned using the full 4K / selected 400 data points from the RL training set, providing a comparison point to assess whether RL training outperforms SFT. (3) **RL on SFT Model**: GRPO is applied to model that has undergone SFT on the selected 400 data points. This allows us to evaluate the impact of initializing RL with a format-aware model, in contrast to starting from the raw instruct model in a cold start manner. (4) **PPO**: We treat the PPO setting as a baseline to evaluate whether our reward design is effective beyond GRPO. We incorporate both the cold start and SFT initialization setting with the same hyper-parameters as GRPO to ensure fairness. Please refer to Appendix E for more details and justifications.

### 3.3 Results

**Main Results.** We report BFCL and API-Bank results in Table 1 and Table 2. Our primary setting trained from scratch on the Qwen2.5-Instruct series generally outperforms other baselines, achieving about 10% absolute gains over SFT trained on the same data volume. In contrast, LLaMA-3.2-Instruct shows less improvement, possibly due to the model's lower adaptability to GRPO-style generalization. Nevertheless, it remains competitive and outperforms most baselines on API-Bank.

---

[2]`https://gorilla.cs.berkeley.edu/blogs/13_bfcl_v3_multi_turn.html`

Table 1: BFCL V3 Benchmark Results, with GRPO cold start as our primary setting.

| Model | Overall Acc | Non-Live AST Acc | Non-Live Exec Acc | Live Acc | Multi Turn Acc | Relevance Detection | Irrelevance Detection |
|---|---|---|---|---|---|---|---|
| Qwen2.5-1.5B-Instruct (**Raw**) | 19.41% | 16.00% | 13.18% | 35.58% | 0.00% | 44.44% | 82.49% |
| Qwen2.5-1.5B-Instruct (**SFT400**) | 40.21% | 65.12% | 61.11% | 56.69% | 1.00% | 94.44% | 60.14% |
| Qwen2.5-1.5B-Instruct (**SFT4k**) | 40.67% | 59.94% | 59.84% | 59.31% | 1.00% | 88.89% | 71.34% |
| Qwen2.5-1.5B-Instruct (**SFT400+PPO**) | 42.95% | 77.65% | 69.75% | 55.73% | 1.88% | 100.00% | 48.40% |
| Qwen2.5-1.5B-Instruct (**SFT400+GRPO**) | 40.93% | 70.54% | 60.79% | 56.33% | 1.00% | 94.44% | 58.63% |
| Qwen2.5-1.5B-Instruct (**PPO Cold Start**) | 38.32% | 79.40% | 70.11% | 45.24% | 0.87% | 100.00% | 18.09% |
| Qwen2.5-1.5B-Instruct (**Ours, GRPO Cold Start**) | **46.20%** | 77.96% | 76.98% | 60.73% | 2.25% | 100.00% | 56.44% |
| Qwen2.5-3B-Instruct (**Raw**) | 33.04% | 42.52% | 40.80% | 53.96% | 1.00% | 64.71% | 56.01% |
| Qwen2.5-3B-Instruct (**SFT400**) | 34.08% | 69.29% | 61.50% | 41.40% | 0.00% | 94.44% | 8.11% |
| Qwen2.5-3B-Instruct (**SFT4k**) | 41.97% | 62.85% | 54.73% | 59.17% | 0.75% | 77.78% | 75.12% |
| Qwen2.5-3B-Instruct (**SFT400+PPO**) | 45.80% | 78.29% | 71.09% | 58.76% | 5.12% | 94.12% | 54.70% |
| Qwen2.5-3B-Instruct (**SFT400+GRPO**) | 46.42% | 76.21% | 68.93% | 64.15% | 1.75% | 88.89% | 71.76% |
| Qwen2.5-3B-Instruct (**PPO Cold Start**) | 51.15% | 82.42% | 78.52% | 67.78% | 4.88% | 94.12% | 73.87% |
| Qwen2.5-3B-Instruct (**Ours, GRPO Cold Start**) | **52.98%** | 81.58% | 79.43% | 73.78% | 3.75% | 88.24% | 84.85% |
| Qwen2.5-7B-Instruct (**Raw**) | 41.97% | 66.02% | 70.11% | 53.51% | 4.25% | 76.47% | 62.66% |
| Qwen2.5-7B-Instruct (**SFT400**) | 34.08% | 69.29% | 66.68% | 41.4% | 0.00% | 94.44% | 8.11% |
| Qwen2.5-7B-Instruct (**SFT4k**) | 36.53% | 45.15% | 53.5% | 57.13% | 0.75% | 72.22% | 72.32% |
| Qwen2.5-7B-Instruct (**SFT400+PPO**) | 42.02% | 83.90% | 72.62% | 51.84% | 0.25% | 100.00% | 29.66% |
| Qwen2.5-7B-Instruct (**SFT400+GRPO**) | 39.25% | 80.69% | 74.34% | 46.51% | 0.25% | 100.00% | 14.19% |
| Qwen2.5-7B-Instruct (**PPO Cold Start**) | 46.68% | 79.33% | 78.16% | 63.17% | 0.38% | 88.89% | 52.92% |
| Qwen2.5-7B-Instruct (**Ours, GRPO Cold Start**) | **58.38%** | 86.17% | 78.25% | 74.9% | 18.12% | 83.33% | 76.68% |
| Llama-3.2-3B-Instruct (**Raw**) | 22.09% | 17.44% | 14.57% | 43.85% | 0.00% | 77.78% | 66.07% |
| Llama-3.2-3B-Instruct (**SFT400**) | 41.22% | 64.27% | 62.18% | 58.37% | 0.75% | 66.67% | 71.12% |
| Llama-3.2-3B-Instruct (**SFT4k**) | **44.16%** | 65.42% | 67.02% | 63.04% | 1.38% | 77.78% | 78.25% |
| Llama-3.2-3B-Instruct (**SFT400+PPO**) | 41.62% | 68.10% | 69.88% | 52.98% | 3.00% | 94.12% | 56.29% |
| Llama-3.2-3B-Instruct (**SFT400+GRPO**) | 42.54% | 65.15% | 68.98% | 59.40% | 0.88% | 72.22% | 65.80% |
| Llama-3.2-3B-Instruct (**PPO Cold Start**) | 42.98% | 84.00% | 72.00% | 52.80% | 2.88% | 100.00% | 31.94% |
| Llama-3.2-3B-Instruct (**Ours, GRPO Cold Start**) | 44.10% | 74.38% | 75.18% | 56.86% | 1.37% | 94.44% | 62.23% |

Table 2: API-Bank Test Results.

| Model | Overall Acc | Level 1 | Level 2 | Level 3 |
|---|---|---|---|---|
| Qwen2.5-1.5B-Instruct (**Raw**) | 30.65% | 28.32% | 35.82% | 35.11% |
| Qwen2.5-1.5B-Instruct (**SFT400**) | 53.60% | 57.14% | 50.75% | 44.27% |
| Qwen2.5-1.5B-Instruct (**SFT4k**) | 47.07% | 52.88% | 52.24% | 26.72% |
| Qwen2.5-1.5B-Instruct (**SFT400+PPO**) | 57.12% | 60.90% | 50.75% | 48.85% |
| Qwen2.5-1.5B-Instruct (**SFT400+GRPO**) | 61.31% | 64.16% | 58.21% | 54.20% |
| Qwen2.5-1.5B-Instruct (**PPO Cold Start**) | 40.54% | 44.61% | 31.34% | 32.82% |
| Qwen2.5-1.5B-Instruct (**Ours, GRPO Cold Start**) | **63.15%** | 70.68% | 61.19% | 41.22% |
| Qwen2.5-3B-Instruct (**Raw**) | 51.59% | 59.65% | 32.84% | 36.64% |
| Qwen2.5-3B-Instruct (**SFT400**) | 52.76% | 59.65% | 50.75% | 32.82% |
| Qwen2.5-3B-Instruct (**SFT4k**) | 50.92% | 55.64% | 43.28% | 40.46% |
| Qwen2.5-3B-Instruct (**SFT400+PPO**) | 65.16% | 67.92% | 55.22% | 61.83% |
| Qwen2.5-3B-Instruct (**SFT400+GRPO**) | 62.48% | 68.67% | 58.21% | 45.80% |
| Qwen2.5-3B-Instruct (**PPO Cold Start**) | 57.62% | 64.66% | 59.70% | 35.11% |
| Qwen2.5-3B-Instruct (**Ours, GRPO Cold Start**) | **67.00%** | 73.43% | 67.16% | 47.33% |
| Qwen2.5-7B-Instruct (**Raw**) | 62.48% | 70.68% | 49.25% | 44.27% |
| Qwen2.5-7B-Instruct (**SFT400**) | 50.59% | 55.89% | 50.75% | 34.35% |
| Qwen2.5-7B-Instruct (**SFT4k**) | 47.07% | 51.13% | 34.33% | 41.22% |
| Qwen2.5-7B-Instruct (**SFT400+PPO**) | 63.15% | 72.43% | 58.21% | 37.40% |
| Qwen2.5-7B-Instruct (**SFT400+GRPO**) | 54.10% | 61.40% | 52.24% | 32.82% |
| Qwen2.5-7B-Instruct (**PPO Cold Start**) | 61.64% | 68.67% | 44.78% | 48.85% |
| Qwen2.5-7B-Instruct (**Ours, GRPO Cold Start**) | **64.66%** | 73.93% | 61.19% | 38.17% |
| Llama-3.2-3B-Instruct (**Raw**) | 40.54% | 44.86% | 29.85% | 32.82% |
| Llama-3.2-3B-Instruct (**SFT400**) | 52.76% | 60.65% | 35.82% | 37.40% |
| Llama-3.2-3B-Instruct (**SFT4k**) | 43.89% | 53.88% | 29.85% | 20.61% |
| Llama-3.2-3B-Instruct (**SFT400+PPO**) | 57.79% | 63.16% | 47.76% | 46.56% |
| Llama-3.2-3B-Instruct (**SFT400+GRPO**) | 56.78% | 63.60% | 41.79% | 43.51% |
| Llama-3.2-3B-Instruct (**PPO Cold Start**) | 55.78% | 60.65% | 41.79% | 48.09% |
| Llama-3.2-3B-Instruct (**Ours, GRPO Cold Start**) | **59.13%** | 65.66% | 52.24% | 42.75% |

Table 3: Bamboogle Test Results

| Model | Accuracy | Avg Num Tool Call |
|---|---|---|
| Qwen2.5-1.5B-Instruct (**Raw**) | 20.8% | 0.61 |
| Qwen2.5-1.5B-Instruct (**SFT400**) | 24.8% | 0.78 |
| Qwen2.5-1.5B-Instruct (**SFT4k**) | 23.2% | 1.25 |
| Qwen2.5-1.5B-Instruct (**SFT400+PPO**) | 36.8% | 1.06 |
| Qwen2.5-1.5B-Instruct (**SFT400+GRPO**) | 38.4% | 0.96 |
| Qwen2.5-1.5B-Instruct (**PPO Cold Start**) | 23.2% | 2.38 |
| Qwen2.5-1.5B-Instruct (**Ours, GRPO Cold Start**) | **44.0%** | 1.19 |
| Qwen2.5-3B-Instruct (**Raw**) | 52.0% | 1.77 |
| Qwen2.5-3B-Instruct (**SFT400**) | 54.4% | 0.86 |
| Qwen2.5-3B-Instruct (**SFT4k**) | 49.6% | 0.92 |
| Qwen2.5-3B-Instruct (**SFT400+PPO**) | 43.2% | 1.04 |
| Qwen2.5-3B-Instruct (**SFT400+GRPO**) | 56.8% | 0.99 |
| Qwen2.5-3B-Instruct (**PPO Cold Start**) | 40.0% | 1.14 |
| Qwen2.5-3B-Instruct (**Ours, GRPO Cold Start**) | **60.0%** | 1.32 |
| Qwen2.5-7B-Instruct (**Raw**) | 69.6% | 1.42 |
| Qwen2.5-7B-Instruct (**SFT400**) | 28.8% | 3.71 |
| Qwen2.5-7B-Instruct (**SFT4k**) | 30.4% | 1.06 |
| Qwen2.5-7B-Instruct (**SFT400+PPO**) | 45.6% | 3.54 |
| Qwen2.5-7B-Instruct (**SFT400+GRPO**) | 29.6% | 3.70 |
| Qwen2.5-7B-Instruct (**PPO Cold Start**) | 48.0% | 1.25 |
| Qwen2.5-7B-Instruct (**Ours, GRPO Cold Start**) | **72.0%** | 1.63 |
| Llama-3.2-3B-Instruct (**Raw**) | 34.4% | 1.25 |
| Llama-3.2-3B-Instruct (**SFT400**) | 44.0% | 0.98 |
| Llama-3.2-3B-Instruct (**SFT4k**) | 48.8% | 0.98 |
| Llama-3.2-3B-Instruct (**SFT400+PPO**) | 39.2% | 1.33 |
| Llama-3.2-3B-Instruct (**SFT400+GRPO**) | 45.6% | 1.00 |
| Llama-3.2-3B-Instruct (**PPO Cold Start**) | 29.6% | 1.42 |
| Llama-3.2-3B-Instruct (**Ours, GRPO Cold Start**) | **52.0%** | 0.89 |

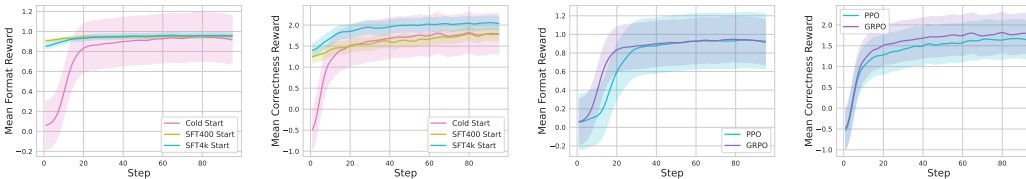

(a) Format Reward (Init.)  (b) Correct Reward (Init.)  (c) Format Reward (Algo.)  (d) Correct Reward (Algo.)

Figure 2: Reward trends across training for Qwen2.5-3B-Instruct. (a)-(b): Different model initialization strategies (Cold Start vs. SFT Initialization). (c)-(d): Different RL strategies (GRPO vs. PPO).

**SFT Initialization Impacts.** Interestingly, GRPO also improves models initialized with limited SFT, often outperforming full-scale SFT trained on 10 times more data. However, this setup still underperforms compared to GRPO cold start. We hypothesize that SFT initialization leads to memorization and overfitting, which reduces the impact of GRPO's effectiveness in generalization. As shown in Figure 2, SFT-initialized models achieve higher training rewards due to distributional alignment between SFT and RL data, but empirically generalize worse on the two benchmarks. This further highlights that higher training rewards do not necessarily translate to better generalization.

**Reward Design on PPO.** We also evaluate PPO under both cold start and SFT-initialized settings to examine the effect of our reward design. The results show that while PPO with a cold start can

Table 4: Cases of Qwen2.5-3B-Instruct (GRPO Cold Start) generalize to irrelevant tool detection.

| User: I would like to buy a movie ticket in San Jose at 11 o'clock in the night.
Tools Available:
1. `Movies_1_BuyMovieTickets(movie_name, number_of_tickets, ...)`
→ Buy tickets for a movie at a specific time and location.
2. `Movies_1_FindMovies(location, ...)`
→ Search movies by location, genre, or format. | User: What is the slope of the line which is perpendicular to the line with the equation y = 3x + 2?
Tools Available:
1. `find_critical_points(function, variable, range=[0.0, 3.4])`
→ Find critical points of a function with respect to a variable, optionally within a range. |
|---|---|
| \<think\> The user wants to buy a movie ticket in San Jose at 11 PM. The available tool, `Movies_1_BuyMovieTickets`, allows us to purchase movie tickets with the required details such as movie name, number of tickets, show date, location, show time, and show type. We need to find the movie name and the show date. However, the user did not specify the movie name or the show date. We can infer the show date from the current date, but we need to know the movie name to proceed with purchase. \</think\> | \<think\> The problem is asking for the slope of a line that is perpendicular to the line with the equation y = 3x + 2. The slope of the given line is 3. The slope of a line that is perpendicular to another line is the negative reciprocal of the original line's slope. So, the slope of the line perpendicular to y = 3x + 2 would be -1/3. The given tool `find_critical_points` is not relevant to finding the slope of a line and cannot be used to solve this problem. \</think\> |
| \<response\> To assist in purchasing a movie ticket, I need the name of the movie and the specific date of the show. Could you please provide this information? \</response\> | \<response\> The slope of the line that is perpendicular to the line with the equation y = 3x + 2 is -1/3. \</response\> |

outperform SFT in some cases, it tends to be less stable across different model settings. In contrast, GRPO consistently achieves higher rewards even from a cold start, suggesting that our reward design works best under the GRPO framework, which contributes to our main experiment setting. As shown in Figure 2, GRPO not only achieves higher correctness rewards but also gains format rewards more rapidly during training. Interestingly, PPO benefits from SFT initialization, generally yielding better results than a cold start, whereas GRPO performs better when trained from scratch. These findings highlight that PPO's gain from our reward design is limited compared to the more robust and consistent improvements observed with GRPO.

**Generalization Studies.** We evaluate the generalization ability of our trained model in two challenging settings: unfamiliar scenarios and novel task goals (both from BFCL benchmark subset). Specifically, we test the model's performance in tool use within unseen programming languages and its ability to detect irrelevant tools, neither of which were explicitly included during RL training or in the dataset. As shown in Figure 3,

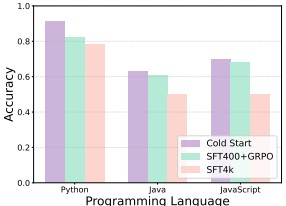
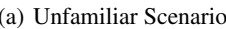
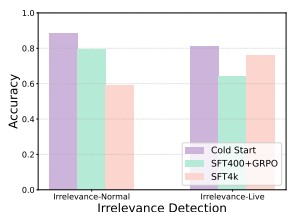

(a) Unfamiliar Scenario  (b) Unfamiliar Goal

Figure 3: Qwen2.5-3B-Instruct's performance across unfamiliar programming languages and novel relevance detection task goals.

Qwen2.5-3B-Instruct, when trained from scratch with our GRPO-based reward design, consistently achieves highest performance. Additionally, Table 4 illustrates two qualitative cases where the model proactively avoids inappropriate tool use: in one, by clarifying an ambiguous intent, and in the other, by answering directly without invoking tools. These behaviors demonstrate emergent proactivity and metacognition, leading to greater efficiency, reduced hallucinations, and signs of agentic intelligence.

**Free-form Inference Effectiveness.** While our model is trained with a focus on tool call format and correctness, we further evaluate its ability to handle free-form tool use in a QA setting. Unlike the structured tool selection and application tasks, QA setting:

Table 5: Performance comparison of our results on Bamboogle vs. Search-R1 under GRPO and PPO training.

| Model | Ours (GRPO) | Ours (PPO) | Search-R1 (GRPO) | Search-R1 (PPO) |
|---|---|---|---|---|
| Qwen2.5-3B | 60.00 | 40.00 | 23.20 | 26.40 |
| Qwen2.5-7B | 72.00 | 48.00 | 40.00 | 36.80 |

(1) imposes no constraints on tool call parameters, and (2) evaluates only the final answer, making it a "goal-oriented" rather than a "process-oriented" task.

Specifically, we use Bamboogle, a multi-hop QA dataset, to assess this capability. The model is equipped with a web search tool, and we report both the answer accuracy and the number of tool calls for all baselines and our approach. As shown in Table 3, our reward design achieves the highest performance, despite this setting not being explicitly seen during training. Notably, our GRPO cold start model surpasses others in accuracy without relying on excessive number of tool calls. This suggests that the model can flexibly invoke tools when needed, effectively leverage feedback, wisely and efficiently navigating toward the correct answer. In addition, to validate the effectiveness of our method, we compared our training result with Search-R1 [17] in Table 5. The results demonstrate that our approach achieves significantly better performance than Search-R1, which further confirms our RL training approach's effectiveness.

## 3.4 Ablation Studies

To validate our experimental design, we conduct ablation studies on three key factors: the scale of RL training data, the relative weighting of correctness rewards, and the effect of including or removing the KL divergence term.

**Ablation on Data Scale.** To investigate the effect of training scale, we conducted ablation experiments by varying the number of RL training examples from 4K to 10K while keeping the distribution, epochs, and GRPO cold-start setting fixed. As shown in Table 6, the performance of Qwen2.5-3B-Instruct on BFCL remains nearly unchanged, with gains within

Table 6: Ablation on training data scale using Qwen2.5-3B-Instruct on BFCL.

| Training Setting | Performance |
|---|---|
| Original 4K RL data | 52.98 |
| Scaling 6K RL data (same distribution) | 53.02 |
| Scaling 10K RL data (same distribution) | 53.31 |

0.5 despite more than doubling the training data. We further observed that the reward convergence curves exhibit nearly identical speeds and final values across all settings. These results suggest that under our reward design, generalization in tool learning is more influenced by reward shaping than the sheer scale of training data. This highlights both the data efficiency of our approach and its practical advantage: strong generalization with limited data, minimizing training cost while maintaining performance. Accordingly, we adopt the 4K dataset as our main RL setting.

**Ablation on Reward Scale.** To justify our choice of relative scaling between format and correctness rewards, we further conduct ablation experiments on BFCL using Qwen2.5-3B-Instruct trained with GRPO. Following Logic-RL [52], we fixed the maximum format reward at 1.0 and varied the scale of the

Table 7: Ablation on correctness reward scale using Qwen2.5-3B-Instruct on BFCL.

| Correctness Reward Scale | 1 | 2 | 3 |
|---|---|---|---|
| BFCL Performance | 40.62 | 51.07 | 52.98 |

correctness reward. As shown in Table 7, setting correctness equal to format (scale = 1) leads to slower convergence and substantially lower performance. Increasing the scale improves performance, with the best result obtained at scale = 3. This indicates that emphasizing correctness more strongly than format is essential for effective RL training in tool use, consistent with prior work where correctness is given higher weight than intermediate format signals. Accordingly, we adopt a correctness reward scale of 3 in our main experiments.

**Ablation on KL Divergence.** We removed the KL penalty across all RL settings, including PPO, to encourage more flexible exploration and better adaptation to our custom tool-use format. As shown in Table 8, the impact on BFCL is minimal, with Qwen2.5-3B showing nearly identical performance

Table 8: Effect of KL penalty on BFCL performance, which shows minimal impact.

| Model | w/o KL | w/ KL |
|---|---|---|
| Qwen2.5-3B-Instruct | 52.98 | 53.05 |
| Qwen2.5-7B-Instruct | 58.38 | 57.21 |

and Qwen2.5-7B slightly improving without KL. Beyond performance, we observed faster convergence, with training reaching stable rewards about five steps earlier, as well as improved efficiency, with total training time reduced by approximately 1.5 times while lowering GPU cost. These findings support our design choice of removing KL in the trainings of the main experiments.

## 4 Analysis

In this section, we perform ablation studies to identify the most effective reward design for tool use, examining reward **type**, **scale**, **granularity**, and **temporal dynamics**. The original (ablated) setting refers to GRPO cold start. All experiments use BFCL benchmark, and we only report overall accuracy for simplicity. Full results could be found in Table 14, Table 15, and Table 16.

### 4.1 Effect of Length Reward

To encourage more elaborate reasoning, we introduce a length-based reward, motivated by prior findings that longer thinking traces can support deeper reasoning [5]. We investigate whether simply promoting longer outputs leads to better task performance in tool use scenarios.

We consider both static and dynamic length rewards, defined as:

$$\mathcal{R}_{\text{length}} = \min\left(\frac{L_{\text{think}}}{L_{\text{target}}}, 1\right), \quad \mathcal{R}_{\text{dynamic}} = \min\left(\frac{L_{\text{think}}}{L_{\text{target}}(1+p)}, 1\right)$$

| (a) BFCL Results (Length) | |
| --- | --- |

| Model | Overall Acc |
| --- | --- |
| Qwen2.5-1.5B-Instruct (**Original**) | **46.20%** |
| Qwen2.5-1.5B-Instruct (**w/ Length Reward**) | 33.23% |
| Qwen2.5-1.5B-Instruct (**Dynamic**) | 28.51% |
| Qwen2.5-3B-Instruct (**Original**) | **52.98%** |
| Qwen2.5-3B-Instruct (**w/ Length Reward**) | 48.89% |
| Qwen2.5-3B-Instruct (**Dynamic**) | 48.24% |
| Llama-3.2-3B-Instruct (**Original**) | 44.10% |
| Llama-3.2-3B-Instruct (**w/ Length Reward**) | **44.98%** |
| Llama-3.2-3B-Instruct (**Dynamic**) | 43.15% |

(b) Response Length

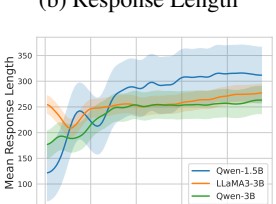

(c) Length Reward

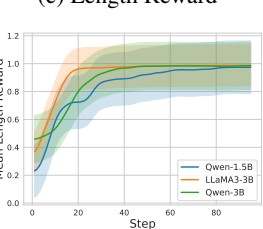

Figure 4: (a) Overall accuracy across models and length reward settings. (b) Response length and (c) corresponding reward trends over training steps under the dynamic length reward setting.

| (a) BFCL Results (Scale) | |
| --- | --- |

| Model | Overall Acc |
| --- | --- |
| Qwen2.5-1.5B-Instruct (**Original**) | **46.20%** |
| Qwen2.5-1.5B-Instruct (**Equal max**) | 39.47% |
| Qwen2.5-1.5B-Instruct (**Two stage**) | 38.85% |
| Qwen2.5-1.5B-Instruct (**Dynamic**) | 45.71% |
| Qwen2.5-3B-Instruct (**Original**) | 52.98% |
| Qwen2.5-3B-Instruct (**Equal max**) | 51.76% |
| Qwen2.5-3B-Instruct (**Two stage**) | 50.66% |
| Qwen2.5-3B-Instruct (**Dynamic**) | **53.81%** |
| Llama-3.2-3B-Instruct (**Original**) | 44.10% |
| Llama-3.2-3B-Instruct (**Equal max**) | 42.47% |
| Llama-3.2-3B-Instruct (**Two stage**) | 41.33% |
| Llama-3.2-3B-Instruct (**Dynamic**) | **46.85%** |

(b) Format Reward        (c) Correctness Reward

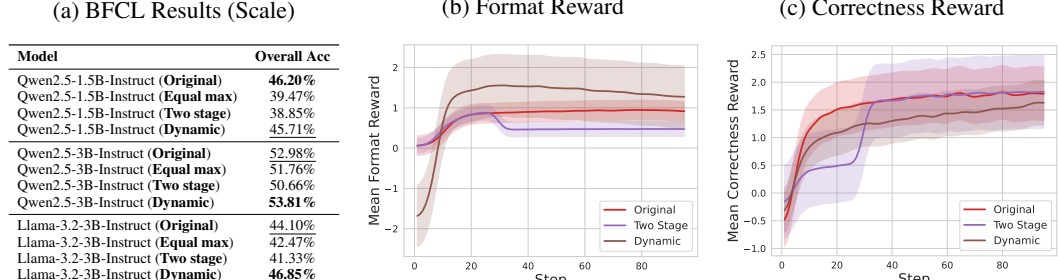

Figure 5: (a) Overall accuracy across models and reward scale settings. (b) Format reward and (c) correctness reward trends across Qwen2.5-3B-Instruct training under different scale dynamics.

where $L_{\text{think}}$ is the length of the <think> segment, $L_{\text{target}}$ is an empirically chosen target length, and $p \in [0, 1]$ denotes normalized training progress. The static reward encourages reasoning up to a fixed target length, while the dynamic reward gradually raises this target as the training progresses.

As shown in Figure 4, both strategies successfully extend reasoning traces. However, the BFCL performance reveal that longer thinking does not consistently improve task success and often degrades performance. This suggests that **while extended reasoning appears beneficial, it can introduce unnecessary complexity, leading to overthinking and reduced effectiveness in tool use**.

## 4.2 Effect of Reward Scale

We investigate how scaling reward components influences learning, particularly the balance between correctness and format rewards. Prior work [52, 17] emphasizes correctness over format to avoid reward hacking, where models exploit superficial patterns without acquiring real task competence.

We test an *Equal Max* variant that equalizes the maximum scales of correctness and format rewards. As shown in Figure 5, this leads to a slight accuracy drop across models, confirming that correctness reward should remain dominant to guide learning toward core reasoning abilities in tool use tasks.

Motivated by the intuition that different learning stages benefit from different focuses, we explore two dynamic scaling strategies: *Two-Stage* approach divides training into two phases: early steps prioritize format learning by downscaling correctness rewards, while later steps reverse this emphasis to focus on correctness; *Dynamic* approach continuously interpolates reward scales throughout training, smoothly shifting focus from format fidelity to correctness as training progresses.

Formal scaling functions are detailed in Appendix H. Figure 5 shows that abrupt shifts in reward emphasis (*Two-Stage*) degrade performance, while gradual adjustments (*Dynamic*) improve it. This suggests that **smoother transitions help models better navigate from simpler objectives to mastering complex reasoning and tool use**.

## 4.3 Effect of Reward Granularity

We finally analyze how the granularity of correctness rewards affects learning. Unlike tasks with definitive answers (e.g., math reasoning), tool use involves multiple facets, making reward assignment more complex. Our original finegrained design decomposes correctness into matching tool names, parameter names, and parameter values, providing a detailed, process-oriented learning signal.

(a) BFCL Results (Granularity)      (b) Correctness Reward Trend

| Model | Overall Acc |
| --- | --- |
| Qwen2.5-1.5B-Instruct (**Original**) | **46.20%** |
| Qwen2.5-1.5B-Instruct (**Finegrained**) | 40.71% |
| Qwen2.5-1.5B-Instruct (**Intermediate**) | 37.65% |
| Qwen2.5-1.5B-Instruct (**Coarse**) | 36.72% |
| Qwen2.5-3B-Instruct (**Original**) | **52.98%** |
| Qwen2.5-3B-Instruct (**Finegrained**) | 52.06% |
| Qwen2.5-3B-Instruct (**Intermediate**) | 51.36% |
| Qwen2.5-3B-Instruct (**Coarse**) | 51.40% |
| Llama-3.2-3B-Instruct (**Original**) | **44.10%** |
| Llama-3.2-3B-Instruct (**Finegrained**) | 39.82% |
| Llama-3.2-3B-Instruct (**Intermediate**) | 38.62% |
| Llama-3.2-3B-Instruct (**Coarse**) | 35.95% |

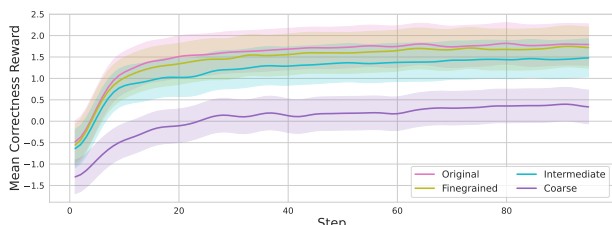

Figure 6: (a) Overall accuracy across models and reward granularity settings. (b) Correctness reward trends across Qwen2.5-3B-Instruct training under different reward granularities.

To assess the impact of granularity, we compare three reward aggregation levels. Relative to the original (most finegrained decomposition) setting: *Finegrained* applies strict match constraints to both tool and parameter names without awarding partial credit; *Intermediate* merges parameter name and value correctness, rewarding only when the entire parameter dictionary matches exactly; *Coarse* treats the entire tool call—including tool name, parameter names, and values—as a single unit, granting reward only for a perfect match. Detailed reward formulations are provided in Appendix I.

As shown in Figure 6, coarser reward formulations result in lower reward attainment and slower learning progress due to sparse and less informative feedback. This highlights that **effective policy optimization benefits not just from stronger rewards but from strategically structured signals that guide the model through complex reasoning processes.**

## 5 Conclusion and Future Work

In this paper, we present a reward design tailored for RL training on tool use tasks. Empirically, our model trained from scratch using GRPO consistently outperforms both SFT-based and SFT-initialized RL baselines across a variety of held-out tool use benchmarks. Furthermore, we demonstrate that our model generalizes well to QA settings, exhibiting robust multi-turn interactions, emergent proactiveness, and metacognitive behaviors, all of which are key traits for efficient and adaptable tool use, lying at the core of foundational agent capabilities.

Our in-depth analysis of reward types, scaling strategies, granularity, and temporal dynamics offers valuable insights into how reward shaping influences both learning efficiency and behavioral outcomes. Building on this foundation, future research could explore: (1) the impact of model scaling in relation to our reward design, (2) the adaptation of our reward framework to embodied agents requiring tool use, and (3) the effectiveness of multi-modal integrated tool use within our proposed ToolRL framework. We hope our findings provide a clear roadmap for advancing the application of reinforcement learning in tool-use scenarios. Ultimately, we envision that *reward is all tool learning needs*, positioning reinforcement learning as a powerful pathway toward developing agents capable of generalizable and creative behaviors.

## Limitations and Broader Impact

While our study highlights the effectiveness of finegrained reward design in enabling tool learning, several limitations remain. First, our experiments primarily focus on policy-gradient methods such as GRPO and PPO, which are naturally compatible with structured, trajectory-level rewards; the generality of our approach to other RL paradigms (e.g., preference-based methods like DPO or SimPO) has not been fully explored. Second, our evaluation is constrained to a limited range of model sizes (up to 7B), leaving open questions about scalability to both smaller lightweight models and much larger foundation models. Third, although our framework demonstrates promising gains in structured reasoning and grounded tool use, it also raises broader social considerations. More capable agents may enhance reliability, interpretability, and user alignment, but the same techniques could be misused if deployed without oversight, particularly in sensitive domains requiring trustworthy decision-making. We view these risks as an important area for ongoing discussion and emphasize the need for responsible application. Overall, our contribution should be seen as an early but timely step toward reward-driven agent training: by dissecting how specific reward components shape learning and generalization, we aim to provide both a practical methodology and conceptual insights that can guide future work in building safe, adaptive, and grounded LLM-based agents.

# Acknowledgement

This research is based upon work supported by DARPA ITM Program No. FA8650-23-C-7316, the AI Research Institutes program by National Science Foundation and the Institute of Education Sciences, U.S. Department of Education through Award # 2229873 - AI Institute for Transforming Education for Children with Speech and Language Processing Challenges. Cheng Qian is supported by a PhD Fellowship from the CapitalOne-Illinois Center for Generative AI Safety, Knowledge Systems, and Cybersecurity (ASKS). The views and conclusions contained herein are those of the authors and should not be interpreted as necessarily representing the official policies, either expressed or implied, of the U.S. Government. The U.S. Government is authorized to reproduce and distribute reprints for governmental purposes notwithstanding any copyright annotation therein.

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

# Appendix

## A  Motivation Details

Training a model's tool selection and application abilities using SFT often suffers from limited generalization. As shown on the left side of Figure 7, a model trained with SFT on deep-thinking trajectories tends to over-interpret tool usage, failing to reject inappropriate tools. Instead of engaging in genuine deep reasoning, it merely imitates superficial cues such as "but wait." This highlights the importance of using reinforcement learning (RL) to guide the model toward more principled decision-making and deeper reasoning capabilities.

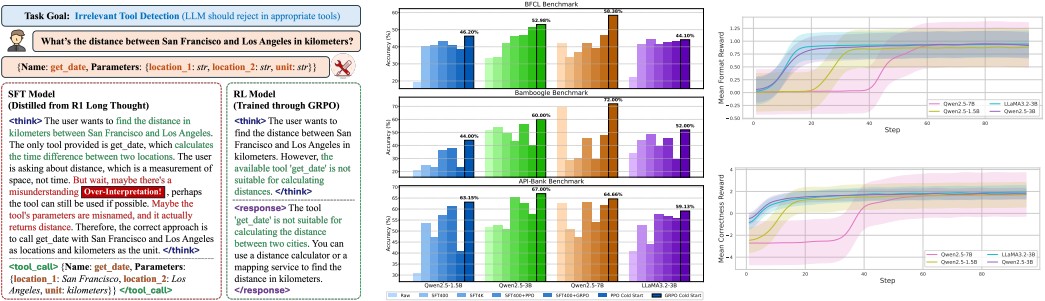

Figure 7: SFT on distilled deep-thinking trajectories leads to overthinking and poor generalization (left). In contrast, RL with our proposed reward design achieves consistently higher performance, with reward curves demonstrating rapid improvement during training (right).

## B  Related Work

**Tool-Integrated Reasoning of LLMs.**    Tool-integrated reasoning (TIR) has emerged as a promising approach to enhance the capabilities of LLMs. Early studies introduced the concept of equipping LLMs with external tools to overcome their inherent limitations [42, 38, 53], such as program executors [6] and search engines [50]. To systematically assess these enhanced capabilities, several benchmarks have been proposed to evaluate tool use performance across various dimensions, including API selection, argument generation, and generalization [39, 29, 33]. Building on this foundation, subsequent research has focused on constructing high-quality tool use datasets [27, 35], enabling models to autonomously create and invoke tools [32, 34], and applying these techniques to problems spanning different modalities [45] and specialized domains [26]. More recently, reinforcement learning (RL) has been explored as an effective framework to further improve TIR, demonstrating success in tasks such as information retrieval [17] and math computation [23]. These advances collectively highlight the growing potential of tool-augmented LLMs for general-purpose reasoning in open-domain settings.

**Exploration of RL in LLMs.**    Previous work has primarily relied on supervised fine-tuning (SFT) with carefully curated datasets to enhance LLM performance in tool use [42, 39]. Recently, reinforcement learning (RL) has gained traction as a more scalable and generalizable training paradigm. The development of RL methods for LLMs has evolved from reinforcement learning from human feedback (RLHF) [19] and proximal policy optimization (PPO) [43] to more advanced techniques such as direct preference optimization (DPO) [40], SimPO [28], and group relative policy optimization (GRPO) [44]. Extensions like dynamic sampling policy optimization (DAPO) [55] and the more recent value-based augmented proximal policy optimization (VAPO) [57] further improve training stability and efficiency.

Among these, GRPO [44] is specifically designed for LLMs, replacing the traditional critic with a group-based evaluation strategy. It has shown strong performance in enhancing reasoning abilities across a range of tasks, including mathematical problem solving [44, 52], search engine interaction [17, 47], and code generation [23]. Beyond task variety, recent studies have analyzed the influence of dataset scale [22] and GRPO's effectiveness in smaller model settings [10]. GRPO's

flexible reward function enables adaptation to diverse objectives, such as assigning weights to sub-tasks [56] or constraining tool use frequency [23]. In this work, we extend GRPO to enhance general tool use capabilities, improving LLMs' ability to select and interact with external tools across a wide range of scenarios.

## C   Algorithm Details

We employ GRPO as our standard RL training setting, a variant of PPO that introduces advantage normalization within grouped samples. This normalization helps stabilize training by reducing variance across samples that share a common input context. We continue to use the symbols defined in Section 2 and further let $\pi_\theta$ represent the current policy.

**Normalized Advantage Across Query Groups.**   For each query $Q$, its responses derived from the rollout form a group $G_Q$ consisting of multiple responses and their corresponding reward values:

$$G_Q = \{A, (s_1, r_1), (s_2, r_2), \ldots, (s_n, r_n)\}$$

where $A$ denotes the ground-truth annotation for $Q$, and each reward $r_i$ is computed as the sum of the format and correctness rewards associated with response $s_i$, i.e., $r_i = \mathcal{R}_{\text{format}}(s_i, A) + \mathcal{R}_{\text{correct}}(s_i, A)$. For each group, we calculate the mean and standard deviation of the rewards:

$$\mu_Q = \frac{1}{n} \sum_{i=1}^{n} r_i, \quad \sigma_Q = \sqrt{\frac{1}{n} \sum_{i=1}^{n} (r_i - \mu_Q)^2}$$

Then, for each sample $s_i$ in the group, we define the normalized advantage:

$$A_i(s_i|Q) = \frac{r_i - \mu_Q}{\sigma_Q + \eta}$$

where $\eta$ is a constant to avoid division by zero.

**Policy Optimization Objective.**   The policy $\pi_\theta$ is optimized using the standard clipped PPO objective, adapted with our group-wise normalized advantages:

$$J_{\text{GRPO}}(\theta) = \mathbb{E}_{Q \sim \mathcal{D}} \mathbb{E}_{s_i \sim \pi_\theta} \Big[ \min \Big( \frac{\pi_\theta(s_i|Q)}{\pi_{\text{old}}(s_i|Q)} A_i(s_i|Q),$$

$$\text{clip} \Big( \frac{\pi_\theta(s_i|Q)}{\pi_{\text{old}}(s_i|Q)}, 1 - \epsilon, 1 + \epsilon \Big) A_i(s_i|Q) \Big) \Big]$$

Overall, this objective guides the policy to generate structurally consistent and semantically accurate tool calls, while group-wise normalization mitigates reward variance across queries, leading to more stable and sample-efficient alignment with task-specific response requirements.

## D   Prompt Details

The **system prompt** we employ during the rollout is shown in Figure 8. The **user prompt** is used to store the trajectory history, including intermediate thoughts, tool calls, environment observations, and any additional user commands. The complete user instruction is presented in Figure 9.

## E   Experiment Details

**Training Data Details.**   We empirically use 4K data points for training, as each dataset consists of samples drawn from the same distribution. Adding more data of similar nature does not increase task diversity. Moreover, we observe that increasing the dataset size beyond 4K does not yield noticeable improvements in the training convergence or final performance, suggesting diminishing returns from additional data under this setting.

Figure 8: The system prompt used for TIR's rollout.

Figure 9: The user prompt used for TIR's rollout.

**GRPO Setting Details.** For all the tool calls in the dataset, we all use JSON format to represent tool call as it's easy to parse and is the most general and structure way of performing tool call. For the GRPO training, we use 2 A100 (80G) GPUs per run with the hyper-parameters shown in Table 9.

**PPO Setting Details.** We apply approximately the same parameter settings as GRPO for the PPO training. Similarly, we use 2 A100 (80G) GPUs per run with the hyper-parameters shown in Table 10.

**Baselines.** The 400 selected data points used for SFT share the same distribution as the 4k data points used for RL training, but differ in content. For SFT, each data point includes a <think> field, with thought content distilled from Deepseek-R1 trajectories. In contrast, GRPO does not require ground truth thought, as only the tool calls are used to compute rewards in the GRPO setting.

We use 400 data points for SFT based on empirical observations that this amount is sufficient to help the raw model learn to follow our tool call format. This provides a stronger initialization and reduces the burden of learning the format from scratch during RL training. However, we also find that relying solely on SFT can lead to overfitting, which may ultimately degrade performance.

Table 9: Full configuration for GRPO training.

| Category | Hyperparameter |
|---|---|
| **Data Configuration** | |
| Train Batch Size | 512 |
| Validation Batch Size | 128 |
| Max Prompt Length | 2048 |
| Max Response Length | 1024 |
| **Optimization** | |
| Learning Rate | 1e-6 |
| PPO Mini Batch Size | 128 |
| KL Loss Used | False |
| **Rollout Configuration** | |
| Rollout Name | vllm |
| GPU Memory Utilization | 0.6 |
| Number of Rollouts | 4 |
| **Training & Logging** | |
| Save Frequency (Steps) | 15 |
| Test Frequency (Steps) | 5 |
| Total Epochs | 15 |

Table 10: Full configuration for PPO training.

| Category | Hyperparameter |
|---|---|
| **Data Configuration** | |
| Train Batch Size | 512 |
| Validation Batch Size | 128 |
| Max Prompt Length | 1024 |
| Max Response Length | 512 |
| **Optimization** | |
| Actor Learning Rate | 1e-6 |
| Critic Learning Rate | 1e-5 |
| PPO Mini Batch Size | 128 |
| PPO Micro Batch Size | 8 |
| **Rollout Configuration** | |
| Rollout Name | vllm |
| GPU Memory Utilization | 0.3 |
| **Training & Logging** | |
| Save Frequency (Steps) | 15 |
| Test Frequency (Steps) | 5 |
| Total Epochs | 15 |

Table 11: BFCL V3 Benchmark Results (Additional Results with 4K SFT Initialization)

| Model | Overall Acc | Non-Live AST Acc | Non-Live Exec Acc | Live Acc | Multi Turn Acc | Relevance Detection | Irrelevance Detection |
|---|---|---|---|---|---|---|---|
| Qwen2.5-1.5B-Instruct (**Raw**) | 19.41% | 16.00% | 13.18% | 35.58% | 0.00% | 44.44% | 82.49% |
| Qwen2.5-1.5B-Instruct (**SFT400+PPO**) | 42.95% | 77.65% | 69.75% | 55.73% | 1.88% | 100.00% | 48.40% |
| Qwen2.5-1.5B-Instruct (**SFT400+GRPO**) | 40.93% | 70.54% | 60.79% | 56.33% | 1.00% | 94.44% | 58.63% |
| Qwen2.5-1.5B-Instruct (**SFT4k+PPO**) | 40.24% | 66.42% | 62.02% | 54.58% | 2.50% | 94.12% | 55.09% |
| Qwen2.5-1.5B-Instruct (**SFT4k+GRPO**) | 42.63% | 66.60% | 64.77% | 60.15% | 1.38% | 88.89% | 67.98% |
| Qwen2.5-3B-Instruct (**Raw**) | 33.04% | 42.52% | 40.80% | 53.96% | 1.00% | 64.71% | 56.01% |
| Qwen2.5-3B-Instruct (**SFT400+PPO**) | 45.80% | 78.29% | 71.09% | 58.76% | 5.12% | 94.12% | 54.70% |
| Qwen2.5-3B-Instruct (**SFT400+GRPO**) | 46.42% | 76.21% | 68.93% | 64.15% | 1.75% | 88.89% | 71.76% |
| Qwen2.5-3B-Instruct (**SFT4k+PPO**) | 48.22% | 77.75% | 73.18% | 64.27% | 5.25% | 94.12% | 66.41% |
| Qwen2.5-3B-Instruct (**SFT4k+GRPO**) | 47.82% | 75.12% | 69.52% | 68.19% | 2.38% | 77.78% | 76.16% |
| Qwen2.5-7B-Instruct (**Raw**) | 41.97% | 66.02% | 70.11% | 53.51% | 4.25% | 76.47% | 62.66% |
| Qwen2.5-7B-Instruct (**SFT400+PPO**) | 42.02% | 83.90% | 72.62% | 51.84% | 0.25% | 100% | 29.66% |
| Qwen2.5-7B-Instruct (**SFT400+GRPO**) | 39.25% | 80.69% | 74.34% | 46.51% | 0.25% | 100% | 14.19% |
| Qwen2.5-7B-Instruct (**SFT4k+PPO**) | 33.80% | 42.67% | 49.50% | 51.80% | 2.38% | 77.78% | 55.79% |
| Qwen2.5-7B-Instruct (**SFT4k+GRPO**) | 35.18% | 43.58% | 50.39% | 55.49% | 0.87% | 77.78% | 67.12% |
| Llama-3.2-3B-Instruct (**Raw**) | 22.09% | 17.44% | 14.57% | 43.85% | 0.00% | 77.78% | 66.07% |
| Llama-3.2-3B-Instruct (**SFT400+PPO**) | 41.62% | 68.10% | 69.88% | 52.98% | 3.00% | 94.12% | 56.29% |
| Llama-3.2-3B-Instruct (**SFT400+GRPO**) | 42.54% | 65.15% | 68.98% | 59.40% | 0.88% | 72.22% | 65.80% |
| Llama-3.2-3B-Instruct (**SFT4k+PPO**) | 45.41% | 73.71% | 68.46% | 62.27% | 2.50% | 82.35% | 68.75% |
| Llama-3.2-3B-Instruct (**SFT4k+GRPO**) | 45.50% | 70.69% | 67.70% | 64.73% | 1.00% | 77.78% | 78.85% |

Table 12: API-Bank Test Results (Additional Results with 4K SFT Initialization)

| Model | Overall Acc | Level 1 | Level 2 | Level 3 |
|---|---|---|---|---|
| Qwen2.5-1.5B-Instruct (**Raw**) | 30.65% | 28.32% | 35.82% | 35.11% |
| Qwen2.5-1.5B-Instruct (**SFT400+PPO**) | 57.12% | 60.9% | 50.75% | 48.85% |
| Qwen2.5-1.5B-Instruct (**SFT400+GRPO**) | 61.31% | 64.16% | 58.21% | 54.20% |
| Qwen2.5-1.5B-Instruct (**SFT4k+PPO**) | 61.31% | 64.91% | 56.72% | 52.67% |
| Qwen2.5-1.5B-Instruct (**SFT4k+GRPO**) | 59.46% | 65.16% | 53.73% | 45.04% |
| Qwen2.5-3B-Instruct (**Raw**) | 51.59% | 59.65% | 32.84% | 36.64% |
| Qwen2.5-3B-Instruct (**SFT400+PPO**) | 65.16% | 67.92% | 55.22% | 61.83% |
| Qwen2.5-3B-Instruct (**SFT400+GRPO**) | 62.48% | 68.67% | 58.21% | 45.80% |
| Qwen2.5-3B-Instruct (**SFT4k+PPO**) | 60.13% | 64.41% | 44.78% | 54.96% |
| Qwen2.5-3B-Instruct (**SFT4k+GRPO**) | 60.80% | 64.41% | 56.72% | 51.91% |
| Qwen2.5-7B-Instruct (**Raw**) | 62.48% | 70.68% | 49.25% | 44.27% |
| Qwen2.5-7B-Instruct (**SFT400+PPO**) | 63.15% | 72.43% | 58.21% | 37.4% |
| Qwen2.5-7B-Instruct (**SFT400+GRPO**) | 54.10% | 61.40% | 52.24% | 32.82% |
| Qwen2.5-7B-Instruct (**SFT4k+PPO**) | 59.30% | 61.40% | 40.30% | 61.60% |
| Qwen2.5-7B-Instruct (**SFT4k+GRPO**) | 52.60% | 56.39% | 34.33% | 50.38% |
| Llama-3.2-3B-Instruct (**Raw**) | 40.54% | 44.86% | 29.85% | 32.82% |
| Llama-3.2-3B-Instruct (**SFT400+PPO**) | 57.79% | 63.16% | 47.76% | 46.56% |
| Llama-3.2-3B-Instruct (**SFT400+GRPO**) | 56.78% | 63.60% | 41.79% | 43.51% |
| Llama-3.2-3B-Instruct (**SFT4k+PPO**) | 54.10% | 60.65% | 40.30% | 41.22% |
| Llama-3.2-3B-Instruct (**SFT4k+GRPO**) | 50.92% | 59.15% | 34.33% | 34.35% |

Table 13: Bamboogle Test Results (Additional Results with 4K SFT Initialization)

| Model | Accuracy | Avg Num Tool Call |
|---|---|---|
| Qwen2.5-1.5B-Instruct (**Raw**) | 20.8% | 0.61 |
| Qwen2.5-1.5B-Instruct (**SFT400+PPO**) | 36.8% | 1.06 |
| Qwen2.5-1.5B-Instruct (**SFT400+GRPO**) | 38.4% | 0.96 |
| Qwen2.5-1.5B-Instruct (**SFT4k+PPO**) | 36.8% | 1.06 |
| Qwen2.5-1.5B-Instruct (**SFT4k+GRPO**) | 34.4% | 1.02 |
| Qwen2.5-3B-Instruct (**Raw**) | 52.0% | 1.77 |
| Qwen2.5-3B-Instruct (**SFT400+PPO**) | 43.2% | 1.04 |
| Qwen2.5-3B-Instruct (**SFT400+GRPO**) | 56.8% | 0.99 |
| Qwen2.5-3B-Instruct (**SFT4k+PPO**) | 46.4% | 1.01 |
| Qwen2.5-3B-Instruct (**SFT4k+GRPO**) | 47.2% | 0.98 |
| Qwen2.5-7B-Instruct (**Raw**) | 69.6% | 1.42 |
| Qwen2.5-7B-Instruct (**SFT400+PPO**) | 45.6% | 3.54 |
| Qwen2.5-7B-Instruct (**SFT400+GRPO**) | 29.6% | 3.70 |
| Qwen2.5-7B-Instruct (**SFT4k+PPO**) | 40.0% | 1.25 |
| Qwen2.5-7B-Instruct (**SFT4k+GRPO**) | 32.0% | 1.25 |
| Llama-3.2-3B-Instruct (**Raw**) | 34.4% | 1.25 |
| Llama-3.2-3B-Instruct (**SFT400+PPO**) | 39.2% | 1.33 |
| Llama-3.2-3B-Instruct (**SFT400+GRPO**) | 45.6% | 1.00 |
| Llama-3.2-3B-Instruct (**SFT4k+PPO**) | 49.6% | 1.02 |
| Llama-3.2-3B-Instruct (**SFT4k+GRPO**) | 42.4% | 1.03 |

# F  Additional Result Details

We present additional results on three benchmarks, applying GRPO and PPO methods to models initialized with SFT on 4K data points. This setting serves as a "theoretical" upper bound, since the same 4K data is first used for SFT and subsequently reused for RL training.

Table 14: BFCL V3 Benchmark Analysis Full Results (Length)

| Model | Overall Acc | Non-Live AST Acc | Non-Live Exec Acc | Live Acc | Multi Turn Acc | Relevance Detection | Irrelevance Detection |
|---|---|---|---|---|---|---|---|
| Qwen2.5-1.5B-Instruct (**Original**) | **46.20%** | 77.96% | 76.98% | 60.73% | 2.25% | 100.00% | 56.44% |
| Qwen2.5-1.5B-Instruct (**w/ Length Reward**) | 33.23% | 70.58% | 71.36% | 35.63% | 0.50% | 94.44% | 4.52% |
| Qwen2.5-1.5B-Instruct (**Dynamic**) | 28.51% | 53.23% | 48.23% | 38.07% | 0.00% | 55.56% | 25.08% |
| Qwen2.5-3B-Instruct (**Original**) | **52.98%** | 81.58% | 79.43% | 73.78% | 3.75% | 88.24% | 84.85% |
| Qwen2.5-3B-Instruct (**w/ Length reward**) | 48.89% | 77.83% | 78.61% | 63.56% | 4.50% | 88.24% | 71.22% |
| Qwen2.5-3B-Instruct (**Dynamic**) | 48.24% | 77.60% | 79.11% | 63.22% | 3.00% | 88.89% | 68.53% |
| Llama-3.2-3B-Instruct (**Original**) | 44.10% | 74.38% | 75.18% | 56.86% | 1.37% | 94.44% | 62.23% |
| Llama-3.2-3B-Instruct (**w/ Length reward**) | **44.98%** | 78.02% | 77.54% | 56.55% | 1.25% | 100.00% | 63.76% |
| Llama-3.2-3B-Instruct (**Dynamic**) | 43.15% | 75.50% | 71.64% | 56.06% | 1.00% | 100.00% | 57.82% |

Table 15: BFCL V3 Benchmark Analysis Full Results (Scale)

| Model | Overall Acc | Non-Live AST Acc | Non-Live Exec Acc | Live Acc | Multi Turn Acc | Relevance Detection | Irrelevance Detection |
|---|---|---|---|---|---|---|---|
| Qwen2.5-1.5B-Instruct (**Original**) | 46.20% | 77.96% | 76.98% | 60.73% | 2.25% | 100.00% | 56.44% |
| Qwen2.5-1.5B-Instruct (**Equal max**) | 39.47% | 78.56% | 75.50% | 45.45% | 2.50% | 100.00% | 16.44% |
| Qwen2.5-1.5B-Instruct (**Two stage**) | 38.85% | 77.96% | 76.23% | 44.51% | 2.25% | 100.00% | 10.61% |
| Qwen2.5-1.5B-Instruct (**Dynamic**) | **45.71%** | 78.31% | 75.73% | 58.91% | 2.50% | 100.00% | 57.20% |
| Qwen2.5-3B-Instruct (**Original**) | 52.98% | 81.58% | 79.43% | 73.78% | 3.75% | 88.24% | 84.85% |
| Qwen2.5-3B-Instruct (**Equal max**) | 51.76% | 81.50% | 79.50% | 69.79% | 4.25% | 88.89% | 78.07% |
| Qwen2.5-3B-Instruct (**Two stage**) | 50.66% | 80.62% | 78.82% | 67.93% | 3.50% | 88.89% | 76.42% |
| Qwen2.5-3B-Instruct (**Dynamic**) | **53.81%** | 81.44% | 80.75% | 75.43% | 3.62% | 77.78% | 88.82% |
| Llama-3.2-3B-Instruct (**Original**) | 44.10% | 74.38% | 75.18% | 56.86% | 1.37% | 94.44% | 62.23% |
| Llama-3.2-3B-Instruct (**Equal max**) | 42.47% | 67.77% | 75.05% | 55.75% | 1.00% | 88.89% | 59.56% |
| Llama-3.2-3B-Instruct (**Two stage**) | 41.33% | 65.54% | 72.70% | 55.22% | 0.75% | 88.89% | 57.59% |
| Llama-3.2-3B-Instruct (**Dynamic**) | **46.85%** | 83.00% | 72.77% | 61.00% | 3.38% | 88.89% | 59.37% |

Table 16: BFCL V3 Benchmark Analysis Full Results (Granularity)

| Model | Overall Acc | Non-Live AST Acc | Non-Live Exec Acc | Live Acc | Multi Turn Acc | Relevance Detection | Irrelevance Detection |
|---|---|---|---|---|---|---|---|
| Qwen2.5-1.5B-Instruct (**Original**) | **46.20%** | 77.96% | 76.98% | 60.73% | 2.25% | 100.00% | 56.44% |
| Qwen2.5-1.5B-Instruct (**Finegrained**) | 40.71% | 78.00% | 75.55% | 48.91% | 2.00% | 100.00% | 24.84% |
| Qwen2.5-1.5B-Instruct (**Intermediate**) | 37.65% | 77.94% | 72.46% | 43.00% | 1.62% | 100.00% | 12.45% |
| Qwen2.5-1.5B-Instruct (**Coarse**) | 36.72% | 76.44% | 70.86% | 41.27% | 2.12% | 100.00% | 12.24% |
| Qwen2.5-3B-Instruct (**Original**) | **52.98%** | 81.58% | 79.43% | 73.78% | 3.75% | 88.24% | 84.85% |
| Qwen2.5-3B-Instruct (**Finegrained**) | 52.06% | 81.65% | 79.64% | 69.21% | 5.50% | 83.33% | 78.14% |
| Qwen2.5-3B-Instruct (**Intermediate**) | 51.36% | 81.15% | 80.07% | 68.64% | 4.25% | 88.89% | 75.74% |
| Qwen2.5-3B-Instruct (**Coarse**) | 51.40% | 79.48% | 78.54% | 68.73% | 5.62% | 88.89% | 77.80% |
| Llama-3.2-3B-Instruct (**Original**) | **44.10%** | 74.38% | 75.18% | 56.86% | 1.37% | 94.44% | 62.23% |
| Llama-3.2-3B-Instruct (**Finegrained**) | 39.82% | 64.71% | 70.68% | 52.20% | 0.25% | 100.00% | 56.68% |
| Llama-3.2-3B-Instruct (**Intermediate**) | 38.62% | 59.83% | 71.86% | 50.56% | 0.25% | 94.44% | 55.68% |
| Llama-3.2-3B-Instruct (**Coarse**) | 35.95% | 52.00% | 61.43% | 48.96% | 1.12% | 83.33% | 61.92% |

The results are shown in Table 11, Table 12, and Table 13 for BFCL, API-Bank, and Bamboogle, respectively. We compare RL training initialized with models fine-tuned on either 400 or 4K SFT data points. Interestingly, our findings suggest that initializing from a model finetuned on 4K data does not consistently outperform initialization from a model finetuned on only 400 data points. In the BFCL benchmark, we even observe cases where performance drops below that of the raw instruct model. This counterintuitive result may stem from overfitting during the SFT phase, which could restrict the model's exploration during RL and lead to poorer generalization on held-out tasks.

# G  Length Reward Analysis Details

We examine the role of a length-based reward. Prior work has demonstrated that the R1-like models can promote deeper reasoning, often reflected in longer thinking traces. To encourage this behavior, we introduce a reward term proportional to the length of the <think> field:

$$\mathcal{R}_{\text{length}} = \min\left(\frac{L_{\text{think}}}{L_{\text{target}}}, 1\right)$$

where $L_{\text{think}}$ denotes the length of the thinking segment in model's output, and $L_{\text{target}}$ denotes the target output length, which we empirically set to 512. We found that the raw model rarely generates responses longer than half this length, making 512 a reasonable and effective target for encouraging longer outputs. This length-based component is added to the overall reward, which now consists of format, correctness, and reasoning length.

As shown in Figure 4, both response length and the length reward generally increase throughout training, particularly for the Qwen model series. This indicates that the length reward effectively encourages longer reasoning. However, the downstream BFCL results reveal that adding a length reward does not consistently improve task performance, and in smaller-scale models, it can even cause substantial degradation. These observations suggest that while extended reasoning may appear

desirable, it is not always beneficial for tool use tasks. In fact, excessive length may introduce unnecessary complexity, leading to overthinking and reduced effectiveness.

**Dynamic Length Reward.** Since fixed-length rewards showed minimal impact and converged quickly, we explored a dynamic length reward that adapts over training steps. Specifically, we define:

$$\mathcal{R}_{\text{dynamic}} = \min\left(\frac{L_{\text{think}}}{L_{\text{target}} \cdot (1+p)}, 1\right)$$

where $S$ denotes the training steps and $p = \frac{S_{\text{current}}}{S_{\text{total}}} \in [0,1]$ represents the normalized training progress. This formulation gradually increases the target thinking length over time, aligning with model maturity.

As shown in Figure 4, this approach yields a steadier growth in thinking length, particularly for the Llama model. However, the BFCL performance results reveal that even scheduled rewards fail to improve performance. This further supports our hypothesis that extended reasoning may not benefit this task and can even have adverse effects.

> 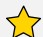 **Takeaway 1**: While length rewards encourage longer reasoning traces, they do not consistently improve task performance and may even harm it in smaller models, highlighting that longer reasoning is not inherently better for tool use tasks.

# H   Reward Scale Analysis Details

We investigate the effect of reward scaling, specifically the relative weighting between correctness and format rewards. Prior work in R1-style RL commonly assigns a higher weight to correctness reward than to format reward, emphasizing the importance of learning correct answer over superficial adherence to format. This strategy helps prevent reward hacking, where a model might exploit formatting heuristics without learning task semantics.

To test the importance of this design choice, we conduct an ablation where we equalize the maximum correctness and format rewards by setting the former's range to $[-1, 1]$, matching that of the format reward. This adjustment only affects the final normalization step of the correctness reward:

$$\mathcal{R}_{\text{correct}} = 2 \cdot \frac{R_{\text{max}}}{S_{\text{max}}} - 1 \in [-1, 1]$$

where all variables are defined as in Section 2.3.

As shown in Figure 5, this equal-scaling variant, denoted as "Equal Max", results in a slight drop in overall accuracy across most models, with the exception of Qwen2.5-3B, which maintains performance comparable to the original setting. These results underscore the importance of assigning greater weight to correctness reward: doing so helps steer the model toward mastering the core reasoning and tool use capabilities necessary for robust generalization.

**Dynamic Reward Scaling.** Building on the insight that correctness reward plays a more critical role, we are further motivated by the intuition that different reward components may benefit from being emphasized at different stages of training. This leads us to explore dynamically adjusting reward scales in accordance with training progress. Specifically, we hypothesize that in early training, the model should prioritize learning the correct output format, which entails an easier objective, before gradually shifting focus to the more challenging goal of tool use correctness. To test this hypothesis, we design two dynamic reward scaling strategies:

- **Two stage (Coarse) Setting**: We divide training into two phases. In the first $s$ training steps, we downscale the correctness reward to $\frac{1}{3}$ of its original scale while keeping the format reward at its original scale. After step $s$, we restore the correctness reward to its original scale and simultaneously reduce the format reward to range $[0, 0.5]$ ($\frac{1}{2}$ of its original scale). Formally the reward scales are:

$$\text{Scale}_{\text{format}} = \begin{cases} [0, 1] & \text{if } S_{\text{current}} < s \\ [0, 0.5] & \text{otherwise} \end{cases},$$

$$\text{Scale}_{\text{correct}} = \begin{cases} [-1, 1] & \text{if } S_{\text{current}} < s \\ [-3, 3] & \text{otherwise} \end{cases}$$

where $S_{\text{current}}$ denotes the current training step. In our experiments, we empirically set the switching point to $s = 30$ steps, as we observed that the format reward typically experiences a significant increase within the first 30 steps. Therefore, it is more beneficial for later steps to shift focus toward optimizing correctness.

- **Dynamic (Finegrained) Setting**: We apply continuous interpolation between the two reward scales throughout training. Initially, both the format and correctness reward scales are set equally. Over time, the format reward scale linearly decays to its original value, while the correctness reward scale gradually increases to its original value, allowing the training to shift focus from format adherence to task correctness accordingly. Formally, the dynamic scaling is defined as:

$$\text{Scale}_{\text{format}} = [-2 + p, 2 - p],$$
$$\text{Scale}_{\text{correct}} = [-2 - p, 2 + p]$$

where $p \in [0, 1]$ similarly represents the normalized training progress. This design ensures a smooth shift of learning focus from format fidelity to correctness.

We present the reward dynamics of the original and two dynamic scaling strategies in Figure 5. The Two-stage (Coarse) reward setting unexpectedly leads to a drop in performance, whereas the Dynamic (Finegrained) scaling could improve model's benchmarking performance. These findings suggest that abrupt shifts in reward scale may negatively impact the training dynamics. In contrast, a smoother and gradual transition from simpler objectives to more nuanced ones appears to better support the model's learning trajectory and generalization during GRPO training.

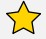 **Takeaway 2**: Gradually adjusting reward scales during training, rather than abrupt changes, better supports model learning and generalization, highlighting the benefits of a smoother transition from simpler objectives to more complex ones.

# I   Reward Granularity Analysis Details

We finally perform a detailed analysis of the effect of reward granularity, focusing specifically on the correctness reward. Tool calling, by nature, poses challenges for reward assignment, as it involves multiple facets beyond a single definitive answer (e.g., in contrast to math reasoning tasks). Our original reward design decomposes correctness into matching the tool name, parameter names, and parameter values, offering a finegrained, "process-oriented" signal that reflects partial correctness in tool usage.

To assess the impact of this granularity, we evaluate three alternative reward formulations with progressively coarser levels of aggregation:

- **Finegrained**: We apply strict exact-match constraints to both tool name and parameter name matching. Specifically, we define:

$$r_{\text{name}} = \mathbb{1}[N_G = N_P] \in \{0, 1\}$$
$$r_{\text{param}} = \sum_{G_j \in G} \mathbb{1}[\text{keys}(P_G) = \text{keys}(P_P)] \in [0, |G|]$$

- **Intermediate**: We combine the parameter name and value rewards into a single term that enforces an exact match on the entire parameter dictionary. Formally:

$$r_{\text{param}} + r_{\text{value}} = \sum_{G_j \in G} \mathbb{1}[P_G = P_P] \in [0, |G|]$$

- **Coarse**: At the coarsest level, we fully entangle tool name, parameter names, and parameter values, treating the entire tool set as a unit. Reward is given only if the generated tool set exactly matches the ground truth:

$$r_{\text{name}} + r_{\text{param}} + r_{\text{value}} = \mathbb{1}[G = P] \in \{0, 1\}$$

All other aspects of reward computation are kept identical to those described in Section 2.3. Starting from our original design, which is the most finegrained, we progressively entangle reward components to derive increasingly coarse-grained alternatives.

The reward dynamics across training steps, shown in Figure 6, demonstrate that as reward granularity becomes coarser, it becomes harder for the model to achieve higher reward values during RL training. This suggests that overly strict and entangled rewards may lead to sparse learning signals, potentially hindering effective credit assignment.

Empirical BFCL test results further support this insight: our original, most finegrained reward strategy performs well across models. In general, finer-grained reward decomposition leads to better training outcomes and higher final task performance, indicating its advantage in promoting more stable and effective policy learning.

> 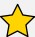 **Takeaway 3**: Finegrained reward decomposition provides richer learning signals, highlighting its role in enabling more effective training compared to coarse reward formulations, which can impede progress and degrade final performance.

## J  Additional Clarifications and Discussions

### J.1  Scope of Our Claim

Our title, *"Reward is All Tool Learning Needs,"* emphasizes the central role of reward design in enabling tool learning within reinforcement learning, rather than claiming the complete absence of supervised signals. We view RL reward itself as a form of supervision, providing structured and interpretable guidance that is particularly well suited to tool-use scenarios. While small-scale supervised examples (e.g., 400–4K) can be used for initial format learning, our main method settings demonstrate that with the proposed format reward, models can acquire correct tool-calling behavior purely through RL training and, in many cases, surpass SFT-only baselines.

Furthermore, our reward design is algorithm-agnostic in principle, though its effectiveness may vary across different RL algorithms. In this paper, we adopt policy-gradient methods such as GRPO and PPO, which naturally accommodate structured, multi-turn rewards and support trajectory-level credit assignment. In contrast, preference-based approaches like DPO or SimPO are less compatible with this setting, as they rely on pairwise comparisons that cannot easily capture finegrained signals such as tool name or argument correctness. Our choices align with prior work (e.g., Search-R1 using GRPO/PPO, ToRL using GRPO) [17, 23], further supporting their methodological suitability.

Finally, while PPO achieves its strongest performance when combined with supervised initialization (e.g., SFT+PPO), our PPO experiments also demonstrate that the proposed reward signals provide meaningful improvements over SFT-only training. Taken together, these findings clarify the scope of our claim: reward design constitutes the foundation of effective tool learning by guiding both format and correctness, whereas supervised data can play a supportive but not fundamental role.

### J.2  Choice of Customized Generation Template

In our experiments, we adopt a customized XML-tag-based generation template instead of relying on the native tool-calling mechanisms of LLMs. Our preliminary comparisons on BFCL with Qwen2.5-3B-Instruct showed that performance under the native tool-call format and our XML-based template was largely comparable. However, the XML design offers several important advantages that motivated our choice. First, it enables structured reasoning through additional fields such as <think>, which improves the model's capacity for explicit intermediate reasoning steps and aligns naturally with agentic paradigms such as ReAct [53], where reasoning and action are explicitly separated. Second, it provides flexibility and extensibility: XML tags can be easily adapted to incorporate new fields, supervision signals, or tool types, which is crucial for generalizing to diverse domains and future research extensions. Third, this approach is consistent with recent works such as Search-R1 [17], ToRL [23], and RM-R1 [7], all of which adopt customized generation formats rather than relying solely on native tool-calls.

While models like LLaMA-3.2-Instruct may exhibit strong zero-shot tool-use performance with native templates due to pretraining exposure, we emphasize that our XML-based template is model-agnostic and optimized for training-time flexibility rather than inference-time execution. Native tool-call format, while convenient, are often rigid and less amenable to reinforcement learning setups that require finegrained, dynamic reward shaping. Our framework prioritizes consistency across models and adaptability to unseen tools and tasks, which we find more critical than exploiting architecture-specific optimizations. In this sense, the XML-based template supports both structured reasoning and broader applicability, making it a better fit for our focus on reward-driven optimization in tool learning.

