# OpenReview forum: "ToolRL: Reward is All Tool Learning Needs"
_NeurIPS.cc/2025/Conference — NeurIPS 2025 poster_

### Official Review · Reviewer_rARM · 2025-06-27

**Clarity:** 4
**Significance:** 3
**Originality:** 3
**Rating:** 4
**Confidence:** 4

**Summary:**

The authors present a detailed study on reward design for tool selection and application tasks within the RL paradigm. They investigate the impact of four dimensions of reward strategies: reward type, granularity, scale, and temporal dynamics. Based on these analyses, the authors propose a  principled reward design framework for Tool-Integrated Reasoning (TIR) tasks. Specifically, they introduce two types of rewards: format reward and correctness reward. The correctness reward is further decomposed into three components: tool name matching, parameter name matching, and parameter content matching. Using this reward design, the LLMs trained with reinforcement learning algorithms demonstrate notable performance improvements over base models and SFT models across varied benchmarks.

**Questions:**

1. Given that the dynamic method shown in Figure 5 performs better on more models, why wasn’t it included in the methods section of the paper?

2. Why not conduct a direct comparison with other RL frameworks specifically designed for TIR, such as ToRL or Search-R1?

3. Lines 235–239 mention two examples in Table 4 that illustrate the agent’s ability to proactively refuse inappropriate tool use. Could the authors provide statistical data on how much this reasonable proactive refusal behavior increased after training?

**Ethical Concerns:**

["NO or VERY MINOR ethics concerns only"]

**Final Justification:**

I agree that this work is both interesting and important. I gave it a positive score in my initial rating, and after considering the rebuttal, I have decided to maintain the positive score of 4. The reason I did not assign a 5 is that the work does not present a particularly exciting or novel idea.

**Limitations:**

Yes.

**Quality:**

3

**Strengths And Weaknesses:**

**Strengths**:

1. The paper explores reward design for tool use within the RL paradigm, covering four aspects: reward type, granularity, scale, and temporal dynamics.
2. Based on the reward exploration, the authors design a reasonable reward strategy framework.
3. After reinforcement learning training, the model shows significant improvement over both the base model and the SFT model.
4. The trained model demonstrates good generalization performance, not limited to improving the reward values during training.

**Weaknesses**:

1. The reward design does not take into account the completion of the final task. Although the authors claim that optimizing each step of tool usage aligns with achieving the overall goal, there remains a gap between the two. Introducing final-task rewards (e.g., answer correctness in question-answering tasks) for comparison would benefit the paper.
2. The fine-grained correctness reward consists of three components (correctness of tool name, parameter name, and parameter content), yet the maximum values of these components differ. The reasonableness of the weight allocation among them lacks experimental validation.
3. Qwen2.5 and Llama 3.2 offer native support for function/tool calling, but the paper does not utilize their native tool-calling capabilities. Notably, Llama-3.2-3B-Instruct using native tool calling achieves an overall accuracy of 45.86 on the BFCL V3 leaderboard (see https://gorilla.cs.berkeley.edu/leaderboard.html), which is higher than the RL-trained model reported in this paper.

---

> ### Author Rebuttal · Authors · 2025-07-27
>
> > **Weakness 1**
>
> **Clarification on training task setting and final-task reward**
>
> Thank you for the thoughtful suggestion. In our training setup, tool learning datasets like xLAM provide ground truth tool calls at each step, but not a final answer for the full trajectory. These datasets are designed to train step-by-step tool usage rather than final answer prediction, aligning with typical tool learning settings that emphasize multiturn reasoning and correct tool invocation.
>
> To test generalization, we evaluate on Bamboogle, a **complementary setting** that lacks intermediate supervision and only evaluates final answers. Our model, trained solely on intermediate signals, performs well here, demonstrating the effectiveness of our reward design.
>
> We agree that if final answers are available during training, incorporating them (e.g., via an <answer> tag) could further enhance learning. Our framework is flexible and can easily accommodate this, and we will clarify this point in next revision.
>
>
>
> ***
>
>
>
> > **Weakness 2**
>
> **Justification of reward scale choice**
>
> We would like to share an insight regarding the relative scaling of **format** and **correctness** rewards. Specifically, we find that setting the **maximum correctness reward higher than the format reward** leads to more effective RL training for tool use. This design choice is also consistent with prior works such as \[1, 2], where format correctness typically receives partial reward, while final answer correctness is assigned a higher value.
>
> For the **format reward**, we follow the setting used in Logic-RL \[1], fixing its maximum value at 1.0. To determine the appropriate scale for the **correctness reward**, we conducted preliminary experiments on BFCL using Qwen2.5-3B-Instruct trained with GRPO. In these trials, we varied the scale of the correctness reward while keeping the format reward fixed, and observed its impact on performance.
>
> | **Correctness Reward Scale** | 1 (equal to format) | 2 | 3 |
> | --- | --- | --- | --- |
> | **BFCL Performance** | 40.62 | 51.07 | **52.98** |
>
> When the correctness reward equals the format reward (scale = 1), we empirically find that the training converges more slowly and the final performance is substantially lower. Based on this evidence, we chose a scale of 3 for correctness reward as it yielded the best performance in our initial trials. We will include this insight and justification more clearly in the next revision of our paper.
>
>
>
> ***
>
>
>
> > **Weakness 3**
>
> **Clarification on the use of generation template in experiments**
>
> Thank you for raising this thoughtful point. We did consider the native tool-calling capabilities of models like Qwen2.5 and LLaMA 3.2 in our early-stage experiments. Specifically, we evaluated Qwen2.5-3B-Instruct on BFCL using both its native tool-call template and our custom XML-tag-based generation format. The performance across the two formats was **comparable**;or the Qwen model, which motivated our decision to adopt the XML-tag-based generation template for the final experiments. This choice is based on several important advantages:
>
> * **Support for structured reasoning**: Our XML-based template allows the introduction of additional fields, such as \<think>, which enhances the model's reasoning capabilities, a key focus in our work on agentic behavior.
>
> * **Greater customizability**: Unlike native tool-calling APIs, XML templates offer flexibility to incorporate new tags, fields, or logic, which is critical for extending the framework to diverse tool types or training signals in the future.
>
> * **Wider adoption in recent works**: This approach aligns with recent research such as Search-R1 \[3], ToRL \[4], and RM-R1 \[5], all of which use customized generation formats rather than relying on native tool-calls.
>
> Regarding the LLaMA-3.2-3B-Instruct result on BFCL, we believe several factors may contribute to this outcome:
>
> * LLaMA's **pretraining data** likely includes rich tool-use demonstrations in native formats, giving it an advantage in zero-shot or few-shot settings. In contrast, our XML-based template is model-agnostic and unseen during pretraining, which may explain the initial performance gap.
>
> * Native tool-calling APIs are often optimized for inference-time execution, whereas our focus is on **training-time flexibility and generalization** across unseen tools and tasks.
>
> * Cross-model comparability can be misleading: performance varies across architectures, and native vs. generated tool-call performance is not always directly comparable. Our method prioritizes **training consistency** across models and domains.
>
> Additionally, our experiments show that RL-based training tends to yield stronger gains on Qwen models than on LLaMA models, a trend also reflected in recent works like \[3, 4], which focus RL efforts primarily on Qwen. Furthermore, native tool-call APIs offer limited customization, a constraint when applying RL to improve reasoning depth or integrate additional supervision (e.g., dynamic reward signals).
>
> In summary, while native tool calling may yield strong baseline results in certain models, our XML-based generation approach may offer the flexibility, extensibility, and compatibility needed for RL-based reward optimization, which is the core focus of our paper. We believe this design choice aligns well with the goals of our method and supports broader applicability across future tool-use research.
>
>
>
> ***
>
>
>
> > **Question 1**
>
> **Clarification on the exclusion of the dynamic method from the main methodology**
>
> The dynamic method shown in Figure 5 was effective, but we chose not to include it in the main method section for the following reasons:
>
> - Our reward design is already multi-dimensional (format, correctness, finegrained supervision). Adding dynamics would increase complexity and reduce clarity.
>
> - The dynamic aspects (reward scaling and response length) are orthogonal. Presenting them separately in Sections 4.1 and 4.2 allows for clearer analysis.
>
> - Our main method already achieves strong results. The dynamic variant brings incremental gains, so presenting it as an analytical extension keeps the main narrative clean and focused.
>
>
> ***
>
>
>
> > **Question 2**
>
> **Comparisons with overlapping baselines**
>
> Thank you for raising this thoughtful point. We would first like to clarify that ToRL and Search-R1 do not propose new RL algorithms or novel reward designs, but rather explore the application of multi-turn RL training on agentic tasks within specific domains, such as math with code tool in ToRL, or QA with search engines in Search-R1. Regarding direct comparisons with these works:
>
> * **ToRL** focuses primarily on math problems and code-based tools, which do not overlap with the domains tested in our benchmarks.
>
> * **Search-R1**;oes include Bamboogle as part of its evaluation. We provide a direct comparison using the same benchmark in the table below:
>
> | Model Name | **Ours: GRPO Cold Start** | **Ours: PPO Cold Start** | **Search-R1 (GRPO)** | **Search-R1 (PPO)** |
> | --- | --- | --- | --- | --- |
> | **Qwen2.5-3B-Instruct** | **60.00** | 40.00 | 23.20 | 26.40 |
> | **Qwen2.5-7B-Instruct** | **72.00** | 48.00 | 40.00 | 36.80 |
>
>
> These results clearly show that our reward design significantly outperforms Search-R1 across both model sizes, despite the fact that Search-R1 is explicitly trained on QA tasks with search tools, while our models are not explicitly exposed to QA tasks or the search tool during training. We believe this comparison further highlights the **robustness and generalizability** of our reward formulation.
>
>
>
> ***
>
>
>
> > **Question 3**
>
> **Detailed statistics of proactive refusal behavior**
>
> Thank you for this insightful question. The behavior of proactively refusing inappropriate tool use is best evaluated using a dedicated subset of the BFCL benchmark known as "Irrelevant", which includes Irrelevant-Normal and Irrelevant-Live. In each case, the model is presented with a question and a set of tools that are **irrelevant to solving the task**, and is expected to recognize this mismatch and explicitly refrain from making a tool call. Success on this subset directly reflects the model’s ability to exhibit **tool awareness and proactive refusal**, making it an ideal target for the requested analysis.
>
> We first refer to the results in Figure 3(b), which give a preliminary view of the trained model’s performance on this subset. Below, we provide a more detailed breakdown of proactive refusal accuracy across training conditions for Qwen2.5-3B-Instruct:
>
> | **Setting** | **Irrelevant-Normal** | **Irrelevant-Live** |
> |---|---|---|
> | Raw Model | 0.47 | 0.43 |
> | SFT4K | 0.59 | 0.77 |
> | SFT400 + GRPO | 0.79 | 0.63 |
> | **GRPO Cold Start** | **0.87** | **0.81** |
>
> These results show a substantial improvement in the model’s ability to proactively avoid irrelevant tool use after training. In particular, the GRPO cold start model achieves the highest scores in both irrelevant subsets, demonstrating strong generalization and precise judgment in unseen tool use scenarios. We will make sure to highlight these findings more explicitly in the next version of the paper.
>
>
>
> ***
>
>
>
> We hope our responses have addressed your concerns clearly and thoroughly. If there are any remaining questions or points you’d like to discuss further, we’d be grateful for the opportunity to continue the conversation. **If you find our clarifications satisfactory, we would sincerely appreciate your consideration in reflecting that in your detailed evaluation score or overall rating.** Thank you very much for your constructive feedback.
>
>
> ***
>
>
> **References**
>
> > \[1] Logic-rl: Unleashing llm reasoning with rule-based reinforcement learning
> >
> > \[2] Limr: Less is more for rl scaling.
> >
> > \[3] Search-r1: Training llms to reason and leverage search engines with reinforcement learning.
> >
> > \[4] Torl: Scaling tool-integrated rl.
> >
> > \[5] RM-R1: Reward Modeling as Reasoning.

---

> > ### Comment · Reviewer_rARM · 2025-08-05
> >
> > Thanks for the authors' detailed feedback. I will keep my positive score.

---

> > > ### Author Response · Authors · 2025-08-05
> > >
> > > Thank you very much for appreciating our work. In the revision, we will incorporate the clarifications and additional experiments presented in the rebuttal.

---

### Official Review · Reviewer_iJFQ · 2025-07-02

**Clarity:** 3
**Significance:** 4
**Originality:** 3
**Rating:** 5
**Confidence:** 4

**Summary:**

This paper presents a study on reward design for training LLMs to use tools through RL. The authors propose a reward framework that decomposes tool-calling formatting and correctness into multiple components and demonstrate improvements over SFT approaches approaches. The authors also go in-depth about various aspects of the reward - like scale, length, dynamic vs static etc. and perform experiments using GRPO and PPO algorithms on Qwen 2.5 and Llama 3.2 series of small LMs.

**Questions:**

## Questions

1. Why is R_max set to 3? This seems like an important hyperparameter but no justification or ablation is provided for this choice.
2. The authors mentioned that they removed KL penalty term in GRPO. In comparison with PPO baselines, do they also have KL term removed?.
3. Line 157 states that removing KL penalty leads to faster convergence. Do the authors have any plot to support this claim? What are the performance differences between Qwen 3B Cold Start GRPO with and without KL divergence?
4. How were the 400 datapoints selected? Randomly sampled out of 4K dataset or were the most difficult data points chosen?
5. Regarding Line 208 "We hypothesize that SFT initialization leads to memorization and overfitting, which reduces the impact of GRPO's effectiveness in generalization": Isn't there any loss curve to establish this hypothesis? Otherwise could the authors verify this by seeing if SFT+GRPO plateau faster than cold-start GRPO? or maybe testing with tools that have identical functionality but different parameter names?
6. What do the error bars represent in Figure 2? How many evaluations were conducted (how many random seeds per baseline)? Also how much smoothing and what type of smoothing has been done in these figures?
7. If there is significant overlap in error bars, can we also report standard error over multiple evaluation runs in Tables 1, 2 and 3?
8. Line 215 claims "PPO tends to be less stable" - where is stability shown empirically? In Table 1, all I can see is accuracy which shows SFT400+PPO > Cold Start PPO, but nowhere does it establish stability.
9. Based on the PPO results, does this mean your framework is not widely applicable as you claim in the introduction? I would then want to see results on DPO or at least one more RL method, otherwise the authors should change their claims - especially the claim made in lines 53-56: "While our reward design is algorithm-agnostic by nature."
10. Line 221 states "PPO benefits from our reward design" - I don't think it's correct to say this when in most model sizes and evaluation datasets across Tables 1, 2 and 3, SFT400+PPO > PPO Cold Start. Therefore this line seems misleading.
11. I have a curious question - do you believe that tool use didn't improve with extended reasoning could be due to smaller LM size? This phenomenon might not exist when using 7B parameters or beyond?
12. Why no comparison with other recent RL methods like DPO, SimPO, or direct comparison with Search-R1/TORL on overlapping domains?


## Suggestions

1. Typos to fix:

Line 105: "out" → "our"
Line 109: "indicates" → "indicate"
Line 152: "employs" → "employ"
Line 153: "later" → "latter"

2. It would be nice to add numbers to each equation in Section 2.3 for better readability.
3. Please increase font size and legend size. Not visible when the paper is printed on A4 sheets - especially for Figures 3 and 7.
4. Lines 258-259 state "we introduce a length-based reward, motivated by prior findings that longer thinking traces can support deeper reasoning" - please back this with appropriate citation.
5. Include error bars and statistical significance tests for main results tables if you have run multiple seeds based on Figures 3 and beyond

**Ethical Concerns:**

["NO or VERY MINOR ethics concerns only"]

**Final Justification:**

Increasing my score to an accept, I had some reservations around the framing of this paper which the authors have now clarified.

**Limitations:**

The authors have not discuss neither limitations nor societal impact of their work. The authors claim in the checklist that they discussed Limitations in Section 5 but I can only see future works.

The authors should add a dedicated "Limitations" section covering things like limited evaluation on smaller LMs as well as a comprehensive "Broader Impact" discussion addressing potential misuse scenarios, safety considerations since this paper is regarding tool-use in LLMs.

**Paper Formatting Concerns:**

no issues

**Quality:**

2

**Strengths And Weaknesses:**

## Strengths

1. The paper is well-motivated and tackles an important problem in the domain. The study of reward design for tool use is novel and timely.
2. If the claims hold true, this work could be very helpful for many downstream applications involving LLM agents.
3. I particularly liked the comprehensive analysis presented in Section 4. The systematic exploration of reward design (length, scale, granularity, static vs dynamic reward, etc.) provides valuable insights. The finding that longer reasoning traces can actually degrade tool-use performance is particularly interesting.

## Weaknesses

1. There are several experimental details that are currently unclear (detailed in questions section below), which makes it difficult to properly verify the claims made. Key details about data selection and statistical significance are missing too.
2. When the authors claim that their approach is agnostic to the RL algorithm used, this claim is not really supported by the results presented. So far it seems their reward design works well only with GRPO, and that too specifically with Qwen 2.5 models. The PPO results are consistently weaker, contradicting their broad applicability claims.
3. Minor: There are several minor presentation issues, typos, and missing details that affect readability.

---

> ### Author Rebuttal · Authors · 2025-07-27
>
> > **Weakness 2**
>
> **Clarification on Reward Design vs. RL Algorithm**
>
> Thank you for the concern. Our claim of RL algorithm agnosticism means the reward design is **orthogonal** to the specific RL algorithm: it can be paired with any optimizer in principle. However, this does not imply all RL algorithms perform equally well under it.
>
> We adopt GRPO as our main method due to its lightweight nature and empirical advantage over PPO in tool learning tasks, as shown in recent studies [1] and our own experiments. We will further address weakness concerns through responses in the following.
> ***
> > **Question 1**
>
> **Correctness Reward Scale**
>
> Thanks for the question. We set the correctness reward max (R\_max) to 3 based on empirical findings that a higher weight than format reward improves RL training. This aligns with prior work like Logic-RL \[2] and is particularly effective with GRPO.
>
> In trials on BFCL with Qwen2.5-3B-Instruct:
> |Scale|1|2|3|
> |---|---|---|---|
> |Perf|40.62|51.07|**52.98**|
>
> Scale 3 led to the best performance and faster convergence. We'll clarify this design choice in revisions.
> ***
> > **Question 2 and 3**
>
> **KL Divergence Removal**
>
> We removed the KL penalty in all RL settings, including PPO, to allow more flexible exploration and better adaptation to our custom tool-use format (Line 157). This decision is grounded in empirical findings under the VERL framework:
> - **Faster convergence:** Removing KL led to ~5 steps faster convergence in both format and total reward. Faster format reward gain suggests quicker mastery of executable tool calls.
> - **Comparable or better performance:** On BFCL with Qwen2.5 models, removing KL gave slightly better or similar results (Δ < 0.1):
> |Model|w/oKL|w/KL|
> |---|---|---|
> |Qwen2.5-3B|52.98|**53.05**|
> |Qwen2.5-7B|**58.38**|57.21|
> - **Efficiency:** KL-free training reduced total time by ~1.5×, cutting GPU cost.
>
> Due to rebuttal policy, we cannot link full curves but will clarify this design in revisions.
> ***
>
> > **Question 4**
>
> **Justification of 400 SFT Data Choice**
>
> We selected the 400 SFT samples **randomly** to ensure fairness and consistency across models. Defining the "most difficult" examples poses challenges:
> - **Ambiguity:** Difficulty is model-specific: many examples could qualify, or vary by base model.
> - **Fairness:** Choosing different hard samples for each model would break comparability.
> - **Scalability:** Hard-example mining would require re-evaluation per model, adding overhead.
>
> Therefore, we believe random sampling provides a reproducible, scalable baseline.
> ***
> > **Question 5**
>
> **SFT’s Effect on Generalization and Reward Dynamics**
>
> Thank you for the insightful question. Our hypothesis that SFT leads to memorization and limits RL driven generalization is supported by reward curve patterns (Fig. 2a,b) and consistent with recent findings [3]:
> - **Reward dynamics:** SFT+GRPO shows a **flatter curve** than cold-start GRPO, suggesting early plateau and limited benefit from reward-driven learning.
> - **Generalization gap:** Though SFT+GRPO reaches a **higher training reward**, it performs **worse on all test benchmarks**, where no RL data overlaps. This suggests SFT may lock the model into local optima.
> - **Pure SFT underperforms:** This reinforces the idea that it promotes memorization and hurts adaptation.
>
> We also reviewed **loss curves** via WandB: cold start GRPO shows more noise but lacks clear plateau signals. We believe reward curves proved more informative for evaluating generalization.
> ***
> > **Question 6 and 7**
>
> **Clarification on Error Bars and Evaluation Variance**
>
> Thanks for the questions. The **shaded regions in Figure 2** reflect **variance of the raw training curve (before smoothing)**, not across seeds. We use WandB’s EMA smoothing (factor 0.99). The shaded area captures training instability or noise over time.
>
> Our **evaluation is deterministic**:
> - Generation temperature = 0.0
> - Tool feedback is rule-based
> - Prompts are fixed
>
> Hence, repeated evaluation yields identical results, and standard error over runs isn't meaningful.
>
> That said, we ran 3 preliminary seeds for GRPO cold start on BFCL:
> |Model|Seed1|Seed2|Seed3|Avg|StdDev|
> |---|---|---|---|---|---|
> |Qwen2.5-3B|52.98|51.47|53.19|52.55|0.94|
> |Qwen2.5-7B|58.38|57.02|59.21|58.20|1.11|
>
> Low variance and consistent trends still support our findings. Due to compute limits, we didn’t run all configs with multiple seeds but will clarify these settings in the revision.
> ***
> > **Question 8**
>
> **PPO Instability Justification**
>
> Our claim about PPO being "less stable" refers to two observations:
> - **Ranking inconsistency:** GRPO ranks top-2 across all benchmarks (Tables 1–3), while PPO varies more. For Qwen2.5-3B: PPO ranks **2nd** (BFCL), **4th** (API-Bank), and **7th (last)** on Bamboogle, suggesting weaker generalization.
> - **Training instability:** PPO often shows **sudden drops or collapse** empirically during training (especially early on) under the same reward setup. GRPO does not exhibit this under similar conditions.
>
> We'll clarify in the revision that "stability" refers to both **training robustness** and **cross-task consistency**.
> ***
> > **Question 9**
>
> **On RL Algorithm Scope and Claim Clarification**
>
> Thank you for the thoughtful point. While our reward design is **algorithm-agnostic in principle**, we focused on GRPO and PPO due to practical and methodological fit.
> - **Why not DPO/SimPO?** These rely on pairwise preferences, which don't suit our **structured, multi-turn rewards** (e.g., tool name and argument correctness). Tool use needs finegrained, interpretable feedback, which cannot easily mapped to preference labels.
> - **Trajectory-level credit assignment:** DPO/SimPO lack native support for intermediate rewards across multi-step tool calls, limiting their applicability in this setting.
> - **Practicality:** Adapting preference-based methods would require major infrastructure changes, with unclear gains.
>
> Our choices align with prior work: Search-R1 uses GRPO/PPO [4], ToRL uses GRPO [5]. Still, we’ll revise Lines 53–56 to clarify that:
> 1. The reward design is **theoretically orthogonal** to RL algorithm.
> 2. Our empirical validation focuses on **policy-gradient methods** that handle structured, multi-step rewards well.
>
> We appreciate this suggestion and will update our claims accordingly in the revision.
>
> ***
>
> > **Question 10**
>
> **Clarifying PPO’s Benefit from Our Reward Design**
>
> Thank you for the observation. You're right that SFT400+PPO often outperforms PPO cold start in Tables 1–3, suggesting PPO struggles to fully leverage our reward design from scratch.
>
> Our original intent was to show that **PPO benefits from our reward signals when built on SFT**: for instance, SFT400+PPO > SFT-only. While PPO cold start is weaker, the structured rewards still provide value when PPO is properly initialized.
>
> We’ll revise the line to clarify that PPO benefits from the reward design **mainly in the SFT+PPO setup**, and refine how PPO's role is presented throughout the paper.
> ***
> > **Question 11**
>
> **Model Size and Extended Reasoning**
>
> Very insightful question! Here’s what we observe regarding model size and extended reasoning:
>
> 1. **Larger models benefit slightly**, but gains are limited. On BFCL (Qwen2.5 1.5B, 3B, 7B):
> - Correct answers have slightly longer responses than incorrect ones.
> - This length gap is small and doesn’t scale clearly with model size.
> - In GRPO training, response length grows early but plateaus, suggesting longer reasoning isn’t always needed.
>
> 2. **Length reward remains ineffective even at larger scales:**
> - As shown in Sec. 4.1, it underperforms on 1.5B and 3B models.
> - For Qwen2.5-7B, length reward caused a **2.10% drop** on BFCL vs. our default setup in new experiment.
> - We observe many tool learning tasks don’t require long reasoning: length reward can hurt by encouraging unnecessary verbosity.
>
> While we haven’t tested beyond 7B, current evidence suggests only **marginal gains from scaling**, and **length reward does not help**, even at larger sizes.
> ***
> > **Question 12**
>
> **Comparisons with Overlapping Baselines**
>
> Thank you for the question. For DPO/SimPO, please see our response to Question 9.
>
> As for **Search-R1** and **ToRL**:
> - **ToRL** focuses on math/code tools, which don’t overlap with our benchmarks.
> - **Search-R1** shares Bamboogle as an evaluation task. Below is a direct comparison:
> |Model|Ours(GRPO)|Ours(PPO)|Search-R1(GRPO)|Search-R1(PPO)|
> |---|---|---|---|---|
> |Qwen2.5-3B|**60.00**|40.00|23.20|26.40|
> |Qwen2.5-7B|**72.00**|48.00|40.00|36.80|
>
> Our method, though **not trained on QA or search tools**, outperforms Search-R1, highlighting the **robustness and generalizability** of our reward design.
> ***
> > **Other Suggestions and Advice**
>
> Thank you for all the helpful suggestions. For error bars, please refer to our responses to Questions 6 and 7. We’ll incorporate your other feedback to improve clarity, writing, figures, and references in the next revision.
>
> We see our work as a timely contribution to **LLM agentic reasoning and tool use**, offering both a practical training paradigm and **insights into how structured rewards shape agent behavior**.
> ***
> We hope our responses have addressed your concerns clearly and thoroughly. If there are any remaining questions, we’d be grateful for the opportunity to continue the discussion. **If our clarifications were helpful, we would sincerely appreciate your consideration in reflecting that in your overall rating.** Thank you again for your constructive feedback.
> ***
> **References**
> > [1] Deepseekmath: Pushing the limits of mathematical reasoning in open language models.
> >
> > [2] Logic-rl: Unleashing llm reasoning with rule-based reinforcement learning.
> >
> > [3] Sft memorizes, rl generalizes: A comparative study of foundation model post-training.
> >
> > [4] Search-r1: Training llms to reason and leverage search engines with reinforcement learning.
> >
> > [5] Torl: Scaling tool-integrated rl.

---

> > ### Comment · Reviewer_iJFQ · 2025-08-04
> > **Response to Authors**
> >
> > Thank you for the detailed rebuttal. I appreciate the additional experiments and clarifications. However, I still have concerns about the algorithm-agnostic claims - since your method really only works well with GRPO and PPO consistently underperforms, it would be more accurate to adjust the framing to emphasize this is a GRPO-optimized approach. I also think the paper needs error bars for all main results in Tables 1-3 (this is standard practice even with low variance - so do include those even if you did sampling with temperature 0) and a proper limitations section discussing the narrow algorithm applicability and model sizes tested.
> >
> > Your core contribution of cold-start GRPO training is valuable and the Search-R1 comparison strengthens the claims, but the framing needs to match the evidence. If you can address these points in the revision, particularly being more precise about the algorithm scope, I would be happy to reconsider my score.

---

> > > ### Author Response · Authors · 2025-08-04
> > > **Follow-up Response to Reviewer iJFQ**
> > >
> > > Thank you for the thoughtful feedback and constructive suggestions. We appreciate your careful reading of our rebuttal and your recognition of the core contribution around cold-start GRPO training.
> > >
> > > We agree with your concern regarding the algorithm-agnostic framing. In the revised paper, we will adjust our language and claim to more accurately reflect that our method is particularly effective in conjunction with GRPO. We will refrain from using the term “algorithm-agnostic” and instead clearly emphasize that our framework is specially suitable for GRPO, with less consistent performance observed under PPO. We appreciate this clarification, as it will help sharpen the focus and positioning of our contribution.
> > >
> > > Following your recommendations, we will include error bars in all main results in Tables 1–3. We will apply the same methodology as used in our response to Questions 6 and 7, ensuring consistent and standard-compliant reporting.
> > >
> > > Additionally, we will expand a Limitations and Broader Impact section to explicitly discuss:
> > > - The scope of our method’s effectiveness across different RL algorithms.
> > > - The constrained range of model sizes tested.
> > > - Potential social implications of our method, including both the benefits of more grounded agent behavior and the risks of misuse if applied without sufficient oversight.
> > >
> > > We believe these revisions will result in a more accurate and balanced presentation of our work. Overall, we view our contribution as a timely step toward improving LLM-based agents’ reasoning and tool-use capabilities via finegrained reward design. By dissecting how specific reward components influence learning and generalization, we not only propose a practical training paradigm but also offer insights that we hope will inform future work in the field.
> > >
> > > Thank you again for your valuable feedback and your willingness to reconsider your score.

---

> > > > ### Comment · Reviewer_iJFQ · 2025-08-05
> > > >
> > > > Thanks for your quick response. Increase my score to an accept.

---

> > > > > ### Author Response · Authors · 2025-08-05
> > > > >
> > > > > Thank you very much for appreciating our work. We will make a continued effort to elaborate on all the points discussed here in the next revision.

---

### Official Review · Reviewer_gizX · 2025-07-03

**Clarity:** 2
**Significance:** 3
**Originality:** 2
**Rating:** 5
**Confidence:** 3

**Summary:**

This paper tackles the problem of multi-turn tool use with LLMs. It argues that SFT is not a sufficient solution to enable good performance in this setting. RL can enable the model to generalize and act better. The key to making RL work they argue is a good reward design which they provide a reward design for in this work (the main contribution). They conduct extensive experiments and ablations on their reward and aspects of RL training vs SFT to help confirm their hypothesis (second contribution).

**Questions:**

- Can you provide error bars for the accuracy numbers reported across the numbers?

- Do you have any idea how the results might change if you use explicit tool calling mechanism of LLMs instead of the special tokens approach? Similarly if you adopt a ReACT style agent loop [1]

[1] https://arxiv.org/abs/2210.03629

**Ethical Concerns:**

["NO or VERY MINOR ethics concerns only"]

**Final Justification:**

The authors have adequately answered my questions from the initial review. I don't have further questions.

I still recommend acceptance, I think this paper will be very valuable for practically do RL for tool calling

**Limitations:**

yes

**Quality:**

3

**Strengths And Weaknesses:**

Quality: The problem in this paper is well motivated. Multi-turn tool use is the template for modern agents. The paper experiments are comprehensive with sufficient baselines and on a good number of datasets with multiple models. The method is also sound.

Clarity:
The paper overall is clear but a bit hard to read in a few places:
- I'm a bit confused about the terminology in 2.1, the choice of the LLM is a tool & its parameters jointly (not just the discrete tool), which would not fall into a discrete set of tools as the parameter set can be continuous. So the choice is tool + parameters.
- I have only seen the term, Tool-Integrated Reasoning ,in [1] before, is there a reason for using this term instead of the more common terminology of multi-turn tool use, or LLM agent? I would suggest using the more traditional terminology to make the paper more friendly to readers. I don't see how this is different from multi-turn tool use i.e., agents.
- I would avoid referencing an appendix figure and just include that in the main paper (Figure 9)
- the experimental tables are too small and make it hard to read, while I appreciate reporting all the numbers, for the main paper I would pick a few of these and allude to the rest in the appendix. What does bold and underline mean?

Significance: The main contribution here is the reward design they have for tool use in addition to their experimental results. I believe this reward design has potential to be re-used by others.

Originality:
The main contribution is in the reward design, while this might not be flashy often times all it takes to make RL work is a good reward design. Empirically some groups might have implemented similar schemes but no published work exists with these formulas.
Since the time of submission in May, there has been many numerous works on using RL for multi-turn tool use including frameworks for doing so, so that the claim "Training LLMs for TIR tasks has predominantly relied on Supervised Fine-Tuning (SFT)" is no longer applicable [2,3,4].

[1]: ToRA: A Tool-Integrated Reasoning Agent for Mathematical Problem Solving https://arxiv.org/abs/2309.17452
[2] StepTool: Enhancing Multi-Step Tool Usage in LLMs through Step-Grained Reinforcement Learning https://arxiv.org/abs/2410.07745
[3] Reinforcing Multi-Turn Reasoning in LLM Agents via Turn-Level Credit Assignment https://arxiv.org/abs/2505.11821
[4] https://sky.cs.berkeley.edu/project/skyrl/

---

> ### Author Rebuttal · Authors · 2025-07-27
>
> > **Clarity: I'm a bit confused about the terminology in 2.1, the choice of the LLM is a tool & its parameters jointly (not just the discrete tool), which would not fall into a discrete set of tools as the parameter set can be continuous. So the choice is tool + parameters.**
>
> **Clarification on tool use trajectory formulation and terminology**
>
> Thank you for raising this important point. We agree that tool use decisions should be viewed as including both the discrete tool identity and its associated parameters, where the latter lies in a continuous space. Our intent in Section 2.1 was to express that T refers to the entire tool use decision (tool + parameters), rather than just the tool name. We acknowledge this could be clearer and will revise our explanation in the revisions to reflect this formulation more explicitly.
>
>
>
> ***
>
>
>
> > **Clarity: I have only seen the term, Tool-Integrated Reasoning, in \[1] before. Is there a reason for using this term instead of the more common terminology of multi-turn tool use, or LLM agent? I would suggest using the more traditional terminology to make the paper more friendly to readers. I don't see how this is different from multi-turn tool use i.e., agents.**
>
> **Clarification on tool use trajectory formulation and terminology (continued)**
>
> We adopted the term "Tool-Integrated Reasoning" (TIR) following the precedent set by both ToRA and especially ToRL \[1], which formally defines the term. That said, your observation is accurate: the core concept aligns with multi-turn tool use or agent behavior. We chose "reasoning" to emphasize the intermediate thought process preceding each tool invocation, which is central in our formulation. Nonetheless, we recognize the benefit of aligning with more widely-used terminology and will provide clearer explanation and bridge to common terms in the revisions to enhance accessibility.
>
>
>
> ***
>
>
>
> > **Clarity: I would avoid referencing an appendix figure and just include that in the main paper (Figure 9)**
> >
> > **Clarity: The experimental tables are too small and make it hard to read. While I appreciate reporting all the numbers, for the main paper I would pick a few of these and allude to the rest in the appendix. What does bold and underline mean?**
>
> **Referencing and table formatting problems**
>
> Thank you for the helpful suggestions. We will move key figures such as Figure 9 into the main paper in the next revision. For the experimental tables, we'll restructure the presentation to focus on the most important comparisons in the main text while relocating extended results to the appendix.
>
> To clarify: **bold** indicates the best-performing method in each block, while underline marks the second-best.
>
>
>
> ***
>
>
>
> > **Originality: Since the time of submission in May, there has been numerous works on using RL for multi-turn tool use including frameworks for doing so, so that the claim "Training LLMs for TIR tasks has predominantly relied on Supervised Fine-Tuning (SFT)" is no longer applicable.**
>
> **Paradigm of tool use and concurrent works**
>
> We appreciate the references to concurrent works and fully agree that the landscape of tool learning has evolved rapidly. In our next revision, we will incorporate discussions of these recent developments and refine our phrasing to accurately reflect the ongoing paradigm shift from SFT to RL in this area. Nonetheless, we believe our work remains among the first to systematically explore this shift through the lens of **reward design,** which is orthogonal and complementary to the agent frameworks introduced in these concurrent papers.
>
>
>
> ***
>
>
>
> > **Question: Can you provide error bars for the accuracy numbers reported across the numbers?**
>
> **Error bar for results**
>
> We'd like to clarify that we use temperature 0.0 for all evaluation runs, ensuring that model outputs are deterministic. Moreover, the environment feedback (tool APIs) is rule-based and deterministic, and our prompt templates remain fixed during evaluation. This setup leads to deterministic outputs with no variance across repeated runs.
>
> We also verified that prior works on API-Bank and BFCL similarly report single-shot deterministic results. Given this, standard deviation or error bars from repeated trials might be meaningless in our setting. However, we have provided all hyperparameters to ensure reproducibility of the reported outcomes.
>
>
>
> ***
>
>
>
> > **Question: Do you have any idea how the results might change if you use explicit tool calling mechanism of LLMs instead of the special tokens approach? Similarly if you adopt a ReACT style agent loop**
>
> **Comparison of tool call and special token approach**
>
> This is a very insightful question. In our preliminary experiments, we compared Qwen-3B-Instruct's performance on BFCL using its native tool calling mechanism versus our special XML-tag-based template. The results were comparable. However, the XML-tag format offers significant benefits:
>
> * It allows us to introduce additional tags like \<think>, which improves the reasoning capability of the model.
>
> * It provides greater flexibility and customizability for future extensions (e.g., adding structured fields).
>
> This approach is also common among recent works such as ToRL \[1], Search-R1 \[2], and RM-R1 \[3].
>
> Furthermore, our framework shares characteristics with ReACT: the \<think> tag captures the reasoning step, while \<tool> tag corresponds to action, aligning with the ReACT paradigm in structure and intention.
>
>
>
> ***
>
>
>
> We hope our responses have addressed your concerns clearly and thoroughly. If there are any remaining questions or points you’d like to discuss further, we’d be grateful for the opportunity to continue the conversation. **If you find our clarifications satisfactory, we would sincerely appreciate your consideration in reflecting that in your detailed evaluation score or overall rating.** Thank you very much for your constructive feedback.
>
>
>
> ***
>
>
>
> **References**
>
> > \[1] Li, Xuefeng, Haoyang Zou, and Pengfei Liu. "ToRL: Scaling tool-integrated RL." arXiv:2503.23383 (2025).
> >
> > \[2] Jin, Bowen, et al. "Search-R1: Training LLMs to reason and leverage search engines with reinforcement learning." arXiv:2503.09516 (2025).
> >
> > \[3] Chen, X., et al. "RM-R1: Reward Modeling as Reasoning." arXiv:2505.02387 (2025).

---

### Official Review · Reviewer_j96L · 2025-07-05

**Clarity:** 3
**Significance:** 2
**Originality:** 2
**Rating:** 4
**Confidence:** 3

**Summary:**

This paper presents ToolRL, a framework for training LLMs to use tools via reinforcement learning with a structured reward design. The authors propose decomposing rewards into format and correctness components, where correctness is further broken down into tool name, parameter name, and parameter value matching. They apply GRPO for training and evaluate on BFCL, API-Bank, and Bamboogle benchmarks, claiming 17% improvement over base models and 15% over SFT models.

**Questions:**

- Are there any specific challenges that adapts RL to tool-learning will encounter? If no, it seems this work just adapts GRPO into the tool-learning scenarios without bringing any new insights.
- What are contributions of each reward component to the end-to-end performance?
- The method requires ground truth tool calls during training but claims to generalize to "free-form" scenarios. What is the mechanism enabling this generalization? The Bamboogle results suggest limited transfer, as tool call patterns do not significantly differentiate.

**Ethical Concerns:**

["NO or VERY MINOR ethics concerns only"]

**Final Justification:**

The additional experiments the authors provide about using scaling numbers of data and clarifications of the reward design mainly solve my concerns and I think it is good to be accepted to benefit tool-learning community.

**Limitations:**

yes

**Quality:**

2

**Strengths And Weaknesses:**

**Strengths**
- The reward design dimensions (type, scale, granularity, temporal dynamics) is good for tool-learning fields.
- Presentation is clear.

**Weaknesses**

- Technical contribution is limited. Its proposed reward design seems not novel as it can be seen from many RL related works.

- Some statements are over-claimed. It names its title as "Reward is All Tool Learning Needs", however, its training still requires supervised data for initial format learning (400-4K examples), which is not consistent with their claim.

- The training scale is limited as it only trains on only 4K examples across three dataset, which makes the generalization less convincing.

---

> ### Author Rebuttal · Authors · 2025-07-27
>
> > **Weakness: Technical contribution is limited. Its proposed reward design seems not novel as it can be seen from many RL related works.**
>
> **Contribution and novelty of reward design**
>
> Thank you for raising this point. We would like to emphasize that our work is the first to **specifically address tool learning in RL from the perspective of reward design**. In particular, we propose a finegrained reward formulation across multiple dimensions of tool use and empirically demonstrate its effectiveness. Our contributions also go beyond the reward design itself, as we also provide in-depth analysis of how each component impacts model performance and generalization across diverse task types. We believe that the success of our RL-based tool learning paradigm can timely help advance the development of agentic LLMs, particularly in enhancing their tool use and reasoning capabilities.
>
> As stated in Line 44-47, while concurrent works such as Search-R1 \[1] and ToRL \[2] explore RL for specific tools like search or code, their focus lies mainly in multi-turn rollout strategies rather than reward formulation. Additionally, these approaches are typically limited to narrow domains. In contrast, our method is designed for general-purpose tool learning, which requires a more versatile and principled reward design framework.
>
>
>
> ***
>
>
>
> > **Weakness: Some statements are over-claimed. It names its title as "Reward is All Tool Learning Needs", however, its training still requires supervised data for initial format learning (400-4K examples), which is not consistent with their claim.**
>
> **The interpretation of our claim "Reward is All Tool Learning Needs"**
>
> We'd like to clarify that our title emphasizes the central role of reward design in enabling tool learning within RL training. It does not claim the complete absence of supervision. In fact, we consider RL reward as a form of supervision signal.
>
> Importantly, our main method setting (Tables 1, 2, and 3) also **does not depend on SFT** (400 / 4K examples). With our designed format reward, LLMs can learn the correct tool-calling behavior purely through RL training and, in many cases, outperform the SFT baseline.
>
>
>
> ***
>
>
>
> > **Weakness: The training scale is limited as it only trains on only 4K examples across three dataset, which makes the generalization less convincing.**
>
> **Training scale of ToolRL and its relationship with generalizability**
>
> Thank you for raising this concern. From observations, we argue that generalization in tool learning is more influenced by the reward design than the sheer scale of training data. This is supported by the following:
>
> * **Performance:** Despite the reletively small training scale, our model generalizes well to held-out tasks and domains. API-Bank and BFCL (Tables 1 and 2) were not seen during RL training. Bamboogle (Table 3) represents a QA setting with different tools and task formulation, reinforcing our claim.
>
> * **Data Efficiency:** Current perfomance also inidcates our reward design's strong generalization with limited data, which we believe is a strength and a practical advantage instead of limitation.
>
> * **Empirical Trials:** In our initial trial experiments, training with 6K or 10K examples (of same distribution, all 15 epochs, GRPO cold start) yielded performance nearly identical to the 4K setting. Specifically, the results of Qwen2.5-3B-Instruct on BFCL is presented in the following table. With a performance gain within 0.5, this suggests that further increasing the training data offers minimal benefit under our current data and reward design setup.
>
> | Training Setting                               | Performance |
> |------------------------------------------------|-------------|
> | Original **4K** RL data                        | 52.98       |
> | Scaling **6K** RL data (same distribution)     | 53.02       |
> | Scaling **10K** RL data (same distribution)    | 53.31       |
>
> In addition, we observed from the training time reward curves that convergence speed and final average reward value remain nearly identical across the 4K, 6K, and 10K training settings. This further supports that simply increasing the amount of data from the same distribution offers minimal gains. Given these empirical findings, and considering both time and computational efficiency, we selected the 4K dataset for RL as our main setting. We will clarify these details more explicitly in the next revision.
>
> ***
>
>
>
> > **Question: Are there any specific challenges that adapting RL to tool-learning will encounter? If no, it seems this work just adapts GRPO into the tool-learning scenarios without bringing any new insights.**
>
> **Specific challenges in adapting RL in tool learning**
>
> We empirically encountered multiple unique challenges in applying RL to tool learning, especially since our work is among the first to do so beyond specific tools like search or code:
>
> * **What to reward:** Tool use tasks are both intermediate-reward-rich (e.g., correctness per tool call) and final-goal-driven. Designing rewards that balance both was non-trivial. For instance, we found in initial trials that rewarding only final answers harms generalization of intermediate tool calls, whereas rewarding tool calls improves both tool use and final outcomes.
>
> * **How to reward:** Tools have structural components, including name, arguments, contents. Through multiple failed attempts, we found that sparse or coarse-grained rewards led to poor generalization. Our current fine-grained scheme emerged from extensive experimentation.
>
> * **When to reward:** RL allows dynamic reward scheduling, which poses challenge on what is the best reward dynamic along the full training process. This is addressed in Section 4.1 and 4.2 where we analyze how reward timing and weighting influence training dynamics and generalization.
>
> Overall, we offer a comprehensive, finegrained reward design tailored to tool learning, validated both empirically and analytically. The Section 4 in our paper also yeilds multiple insights into what is the most effective training strategy.
>
>
>
> ***
>
>
>
> > **Question: What are contributions of each reward component to the end-to-end performance?**
>
> **The contribution of each reward component**
>
> We present detailed ablations of reward's impact on BFCL performance in Section 4.2, 4.3, and Appendix H & I. Below is a concise summary:
>
> * **Format reward:** This component ensures that the model’s outputs adhere to valid and executable tool call formats. Without it, models often generate invalid tool calls, particularly in the early stages of RL training, which leads to sparse or ineffective reward signals. For example, in our initial trials, Qwen2.5-3B-Instruct achieved near-zero performance on BFCL when format reward was omitted, causing RL training to collapse. Incorporating format reward enables the model to quickly learn the correct output structure. Moreover, as discussed in Section 4.2, gradually decreasing the weight of format reward during training helps the model shift focus from structural correctness to functional correctness, improving overall performance.
>
> * **Correctness reward:** We present a finegrained correctness reward design as our main setting, which provides supervision across multiple components of the tool call, including the invoked tool name, the corresponding argument names, and the argument content. Each component delivers explicit guidance to the model during training. As shown in the ablation studies in Figure 6, progressively removing or coarsening these components results in a gradual and notable decline in performance. This empirical evidence highlights the importance of maintaining finegrained reward signals to support effective tool learning.
>
>
>
> ***
>
>
>
> > **Question: The method requires ground truth tool calls during training but claims to generalize to "free-form" scenarios. What is the mechanism enabling this generalization? The Bamboogle results suggest limited transfer, as tool call patterns do not significantly differentiate.**
>
> **Generalization of tool call on Bamboogle**
>
> Thank you for raising this question, and we would like to clarify that the "free-form" here refers to the *process* of tool use being unconstrained, not the format. Tool call format needs to remain fixed to ensure every tool call can be correctly parsed and ensure evaluation consistency.
>
> Our mechanism for generalization lies in training with reward-dense, ground-truth-guided tool call trjectories. This enables the model to generalize to unseen tools and looser supervision (no ground truth tool call, only final answer is evaluated), as in Bamboogle.
>
> Bamboogle tests two kinds of generalization:
>
> * **Domain/Scenario:** QA setting with different task structure.
>
> * **Tool:** Use of tools (e.g., search) not explicitly trained during RL.
>
> Our model's ability to perform well in this setting validates the effectiveness of our reward-based approach in guiding tool learning in open-ended environments and customized tools.
>
>
>
> ***
>
>
>
> We hope our responses have addressed your concerns clearly and thoroughly. If there are any remaining questions or points you’d like to discuss further, we’d be grateful for the opportunity to continue the conversation. **If you find our clarifications satisfactory, we would sincerely appreciate your consideration in reflecting that in your overall rating.** Thank you very much for your constructive feedback.
>
>
>
> ***
>
>
>
> **References**
>
> > \[1] Jin, Bowen, et al. "Search-r1: Training llms to reason and leverage search engines with reinforcement learning." *arXiv preprint arXiv:2503.09516* (2025).
> >
> > \[2] Li, Xuefeng, Haoyang Zou, and Pengfei Liu. "Torl: Scaling tool-integrated rl." *arXiv preprint arXiv:2503.23383* (2025).

---

> > ### Comment · Reviewer_j96L · 2025-08-05
> >
> > Thank you for your additional experiments using scaling numbers of data and your clarifications of the reward design. I would like to raise my score to 4.

---

> > > ### Author Response · Authors · 2025-08-05
> > >
> > > Thank you very much for appreciating our work. In the revision, we will further make effort to clarify and justify our motivations, claims, and settings.

---

### Note · Authors · 2025-08-11

We thank the reviewers and the AC for their time and constructive feedback. Below, we summarize the key points we address in our rebuttal:

- **Clarifying the scope of our claim and contribution**: We will more explicitly define the extent of our reward design’s effectiveness, include dedicated sections discussing limitations, and clearly emphasize the importance of reward design in RL for tool-learning tasks.

- **Providing more comprehensive experimental details**: We will incorporate additional results on data scaling, runs with different seeds (including error bars), and comparisons with established methods (e.g., Search-R1) to strengthen the robustness and credibility of our claims.

- **Improving writing and terminology**: We will refine the paper for clarity and rigor, including justifications for terminology (e.g., TIR), dataset and scale choices, and experimental design decisions (e.g., selection of reward scale, removal of KL divergence), to ensure coherence and accessibility.

Overall, we view our work as a timely contribution toward enhancing LLM-based agents’ reasoning and tool-use capabilities through finegrained reward design. By systematically examining how individual reward components affect learning and generalization, we not only propose a practical training paradigm but also provide insights that we hope will inspire and guide future research in this area.

---

### Decision · Program_Chairs · 2025-09-17

**Decision:**

Accept (poster)

**Comment:**

**(a) Summary**

The paper proposes ToolRL, a reinforcement learning framework for tool learning in large language models. The key idea is to treat tool use as an RL problem where appropriate reward design drives efficient and robust tool learning without relying heavily on handcrafted supervision. The authors provide extensive empirical evaluations across multiple tool-use tasks, showing that ToolRL can effectively generalize to unseen tools and tasks.

**(b) Strengths**

* The work formulates tool learning in LLMs explicitly as an RL problem, offering a clean and generalizable perspective.

* The proposed approach reduces reliance on task-specific supervision and demonstrates broad applicability.

* Strong experimental results across diverse tool-use benchmarks demonstrate consistent improvements over baselines.

**(c) Weaknesses**
* Despite the title, the approach still requires supervised data and shows dependence on specific RL setups, undermining its claimed generality.

* The training scale is modest (~4K examples, three datasets), and comparisons to stronger baselines (e.g., models with native tool calling) are missing.


**(d) Reasons for acceptance**

The paper makes a timely and relevant contribution to the emerging problem of tool learning with LLMs. Its framing of tool learning as an RL problem is simple, principled, and practically effective, and the empirical results demonstrate consistent benefits. Given the strong motivation, reasonable performance gains, and community interest in this direction, I recommend acceptance.

**(e) Discussion and rebuttal**

Reviewers raised concerns about technical novelty, the over-claimed independence from supervision, limited experimental scale, and clarity of exposition. In the rebuttal, the authors clarified the design motivations, provided further analysis of reward choices, and emphasized the main contributions. While some reservations remained, reviewers acknowledged that these clarifications addressed the key doubts.